# Identifiable Smooth Conjugacy Learning via Adversarial Orthogonality

**In Huh** [1 2]   **Changwook Jeong** [3 4]   **Muhammad Ashraful Alam** [1]

## Abstract

Data-driven dynamical system models often fail to recover the long-term structure of the underlying system, as their behavior is weakly constrained off the data manifold. Conjugacy-based approaches address this limitation by learning a diffeomorphism that pushes forward a source vector field to match observed dynamics, inheriting qualitative topology from the source. However, such methods typically presuppose that the chosen source system is topologically compatible with the target data. When this assumption is violated, the conjugacy problem becomes ill-posed, and arbitrary corrections can be traded off against diffeomorphic variation, leading to non-identifiability. We propose a framework that relaxes this assumed prior by jointly learning the diffeomorphic conjugacy together with controlled adjustments to the source dynamics via low-dimensional context modulation. Inspired by versal unfolding theory, we enforce the modulation space to be orthogonal to the worst-case orbit-tangent directions, obtained by adversarially searching over a class of parameterized diffeomorphisms. This promotes an identifiable decomposition of dynamical variation into diffeomorphic and intrinsic, topology-changing components, enabling interpretable corrections that recover the canonical structure such as normal forms and symmetries.

## 1. Introduction

Dynamical systems modeling forms a foundational pillar of systematic understanding across scientific and engineering disciplines. Recently, data-driven approaches that learn dynamical systems directly from observations (Brenner et al., 2025; Brunton et al., 2016; Chen et al., 2018; Kirchmeyer et al., 2022; Nzoyem et al., 2025; Yin et al., 2021a;b) have emerged as powerful complements to expert-driven modeling, enabling dynamical analysis in domains where first-principles knowledge is incomplete or unavailable.

Crucially, learning a dynamical system is not merely a matter of fitting short-term rollouts. A successful data-driven model should also faithfully reproduce the *topological invariants* of the target system, including long-term statistics and topological properties of invariant sets, such as the type, number, and indices of equilibria (Durstewitz et al., 2026; Huh et al., 2025; Park et al., 2024). In this sense, an ideal reconstructed model should be *topologically equivalent* to the true system, generating trajectories that preserve the same qualitative structure beyond the training horizon.

A major obstacle to this goal is the Out-Of-Domain (OOD) (Göring et al., 2024; Huh et al., 2025) problem. In many realistic settings, data populate only a small subset of the phase space, often confined to a single dynamical regime. Scientific use, however, frequently demands extrapolation across qualitatively different regimes—especially near bifurcations or basin boundaries, where invariant sets and stability types change (Kuznetsov, 1998). In such settings, black-box models without adequate structural priors tend to generalize only locally around observed region, while their behavior away from the data manifold may become arbitrary and fail to preserve long-term invariants.

This motivates a line of approaches that inject structural priors to constrain the hypothesis class to models that preserve qualitative dynamical properties. One principled route is *conjugacy*: two dynamical systems are said to be *smoothly conjugate* if there exists a diffeomorphism that transforms one system into the other. Such systems are necessarily topologically equivalent and share the same qualitative dynamics. Building on this idea, recent methods (Friedman et al., 2025; Moriel et al., 2024; Sagodi & Park, 2025) propose learning a diffeomorphism that pushes forward a known source dynamics to match observed target data, thereby inheriting the qualitative structure of the source system by construction.

Despite their appeal, such frameworks rely on a strong prior assumption: the chosen source system must already be con-

[1] Elmore Family School of Electrical and Computer Engineering, Purdue University [2] CSE Team, Samsung Electronics [3] Graduate School of Semiconductor Materials and Devices Engineering, UNIST [4] Artificial Intelligence Graduate School, UNIST. Correspondence to: In Huh <ihuh@purdue.edu>, Changwook Jeong <changwook.jeong@unist.ac.kr>, Muhammad Ashraful Alam <alam@purdue.edu>.

*Proceedings of the 43rd International Conference on Machine Learning*, Seoul, South Korea. PMLR 306, 2026. Copyright 2026 by the author(s).

jugate to the data-generating system, at least within the regime of interest. When this topological compatibility fails, the conjugacy problem becomes ill-posed and learning may break down. On the other hand, simply relaxing the source prior by allowing learnable corrections can introduce a different failure mode: the correction may represent essentially arbitrary dynamical variation, entangling intrinsic topology-changing effects with diffeomorphic reparameterizations and yielding non-identifiable solutions. This ambiguity can produce corrections that alter qualitative topology or disrupt canonical structures such as symmetries, limiting interpretability and scientific applicability.

In this work, we propose a learning framework that relaxes the need for the source system to encode a strong topological prior, while retaining the key advantages of conjugacy-based modeling. Motivated by *unfolding theory* (Arnold, 2012), we allow structured modifications of the source dynamics, but restrict them to remain controlled and identifiable. Concretely, we jointly train a diffeomorphic conjugacy together with *low-dimensional, context-driven adjustments* (Kirchmeyer et al., 2022; Nzoyem et al., 2025) to the source vector field. Importantly, we impose a geometric constraint that encourages these context-induced variations to be *orthogonal to worst-case orbit-tangent directions*, obtained by adversarially searching over a parameterized family of diffeomorphisms. This orthogonality promotes a minimal yet sufficient correction to capture meaningful perturbations of the source dynamics, preventing diffeomorphic and intrinsic modifications from collapsing into entangled representations. As a result, our framework admits an identifiable decomposition of source-target discrepancy into diffeomorphic and topology-changing components, recovering the canonical structure of the underlying system.

## 2. Preliminary

**Smooth conjugacy.** Let $\mathcal{X} \subseteq \mathbb{R}^{d_x}$ be an open set, and consider two autonomous ODEs on $\mathcal{X}$:

$$\dot{x} = f(x), \qquad \dot{y} = g(y), \qquad x, y \in \mathcal{X},$$

where $f, g \in \mathfrak{X}(\mathcal{X})$ are vector fields. Let $\varphi_f, \varphi_g : \mathcal{X} \times \mathbb{R} \to \mathcal{X}$ denote the flows generated by $f$ and $g$, and write $\varphi_f^\tau(x) = \varphi_f(x, \tau)$. We say that $f$ and $g$ are *smoothly conjugate* if there exists a diffeomorphism (a smooth and invertible mapping) $\Phi : \mathcal{X} \to \mathcal{X}$ such that

$$\varphi_g^\tau = \Phi \circ \varphi_f^\tau \circ \Phi^{-1}, \tag{1}$$

for all $\tau \in \mathbb{R}$ for which both sides are defined. Equivalently, $g$ is the pushforward of $f$ by $\Phi$:

$$g = \Phi_* f, \qquad (\Phi_* f)(y) := D\Phi(x) f(x)\big|_{x=\Phi^{-1}(y)}, \quad (2)$$

where $D\Phi(x)$ denotes the Jacobian of $\Phi$ at $x$. Indeed, differentiating the flow-level conjugacy (1) with respect to $\tau$ at

$\tau = 0$ yields the infinitesimal condition (2), and conversely, the latter integrates to the flow conjugacy by uniqueness of solutions. These relations establish the dynamical equivalence between $f$ and $g$, ensuring that their qualitative phase portraits and topological invariants coincide.

**Conjugacy learning.** Building on the notion of smooth conjugacy, recent works (Friedman et al., 2025; Sagodi & Park, 2025) propose to learn a diffeomorphism, typically modeled by $\theta$-parameterized normalizing flows (Dinh et al., 2017; Papamakarios et al., 2021) $\Phi_\theta$, that pushes a source vector field $\dot{x} = f(x)$ to a target dynamics $\dot{y} = g(y)$, under the prior assumption that $f$ is smoothly conjugate to $g$. Here, $f$ may be either a known symbolic model or a Neural Ordinary Differential Equation (NODE) (Chen et al., 2018) trained on abundant, low-cost data, whereas $g$ is unknown and accessible only through sparse measurements.

When paired data $\{(y_n, \dot{y}_n)\}_{n=1}^N$ are available, this can be achieved by minimizing the infinitesimal conjugacy loss

$$\mathcal{L}_{\text{conj}}^{\text{vf}} = \frac{1}{N} \sum_{n=1}^N \left\| (\Phi_\theta)_* f(y_n) - \dot{y}_n \right\|^2 \tag{3}$$

that encourages smooth conjugacy through (2). When only trajectory data $\{Y_n\}_{n=1}^N$, with $Y_n = \{y_n^0, ..., y_n^T\}$, are available, one instead minimizes a flow-level conjugacy loss:

$$\mathcal{L}_{\text{conj}}^{\text{flow}} = \frac{1}{NT} \sum_{n=1}^N \sum_{t=1}^T \left\| y_n^t - \Phi_\theta(\varphi_f^{t\Delta\tau}(\Phi_\theta^{-1}(y_n^0))) \right\|^2, \tag{4}$$

where $\Delta\tau$ denotes the time interval. It encourages smooth conjugacy at the level of flows as a form of $\varphi_g^\tau(y(0)) = y(\tau) = \Phi \circ \varphi_f^\tau \circ \Phi^{-1}(y(0))$, corresponding to (1). Practically, conjugacy losses are optimized together with complexity regularization on $\Phi_\theta$, such as log-volume penalties $\| \log \det D\Phi_\theta \|^2$ or near-identity constraints $\|\Phi_\theta(x) - x\|^2$. In this work, we mainly use (4), as it aligns with the standard trajectory-only supervision in dynamical systems modeling.

Both (3) and (4) enforce that the transformed source system reproduces the observed target dynamics under smooth conjugacy. In other words, the diffeomorphism $\Phi(\cdot; \theta)$ is trained to align the pushforward of the source dynamics with the target observations, while preserving the underlying dynamical structure of the source system by construction.

Consequently, when the source and target systems indeed belong to the same smooth conjugacy class, this formulation guarantees recovery of the correct topological structure of the target dynamics, even from sparsely observed data. If, however, a topological mismatch exists, the conjugacy assumption is violated at a structural level: no diffeomorphism can bridge the two systems, and the learning problem becomes intrinsically ill-posed, precluding faithful reconstruction of the target dynamics.

## 3. Proposed Method

### 3.1. Topological Correction via Low-rank Modulation

**Mismatch correction.** Relaxing the assumption of topological compatibility between the source and target systems, we introduce a learnable corrective vector field $\delta_\xi \in \mathfrak{X}(\mathcal{X})$, parameterized by $\xi \in \mathbb{R}^{d_\xi}$, to compensate for the topological mismatch. We then model the target dynamics as

$$g = \Phi_* f_\xi = \Phi_*(f_0 + \delta_\xi), \qquad \varphi_g^\tau = \Phi \circ \varphi_{f_\xi}^\tau \circ \Phi^{-1}, \quad (5)$$

where $f_\xi := f_0 + \delta_\xi$ denotes an adjusted source dynamics, and $f_0$ is a prespecified source system that is not necessarily smoothly conjugate to the target system $g$. Accordingly, the conjugacy loss in (4) is extended to a joint objective over both the diffeomorphic map and the corrective dynamics:

$$\mathcal{L}_{\text{conj}}^{\text{flow}} = \frac{1}{NT} \sum_{n=1}^N \sum_{t=1}^T \left\| y_n^t - \Phi_\theta(\varphi_{f_\xi}^{t\Delta\tau}(\Phi_\theta^{-1}(y_n^0))) \right\|^2, \quad (6)$$

which is minimized with respect to both $\theta$ and $\xi$.

**Low-rank context modulation for correction.** There is flexibility in how the correction term $\delta_\xi$ is parameterized. In principle, $\delta_\xi$ may be realized as a fully expressive neural network, or endowed with additional structural bias. Among these possibilities, we adopt context-informed low-rank weight modulation, recently introduced in NODE literature (Kirchmeyer et al., 2022; Nzoyem et al., 2025). This choice is particularly well suited for capturing small yet topologically consequential modifications of vector fields: topological transitions in dynamical systems are typically governed by low-dimensional parameter families (Kuznetsov, 1998), even when the state space is high-dimensional.[1]

Formally, let $f_0(x) := f(x; \phi_0)$ denote the reference NODE vector field with the fixed weight parameters $\phi_0 \in \mathbb{R}^{d_\phi}$. We parameterize a structured corrective perturbation via a low-rank modulation of the weights:

$$f_\xi(x) := f(x; \phi_0 + W\xi), \quad (7)$$

where $\xi \in \mathbb{R}^{d_\xi}$ now plays a role of a low-dimensional context vector rather than an arbitrary parameterization, i.e., $d_\xi \ll d_\phi$. $W \in \mathbb{R}^{d_\psi \times d_\xi}$ maps this low-dimensional context vector to the full weight space. By construction, $\xi = 0$ recovers the reference vector field $f_0(x)$, and the modulated weights $\phi = \phi_0 + W\xi$ lies in an affine subspace of dimension $\text{rank}(W)$ within the ambient weight space.

In practice, the modulation $W\xi$ is learned under complexity control, such as norm or sparsity regularization, so that the context $\xi$ remains confined to a small neighborhood of the

---

[1]This regime is commonly referred to as a finite-codimension bifurcation (Kuznetsov, 1998).

origin $\phi_0$ (Kirchmeyer et al., 2022). Under this regime, a first-order expansion of (7) around $\phi_0$ yields

$$f_\xi(x) = f_0(x) + \sum_{i=1}^{d_\xi} \xi_i p_i(x) + O(\|\xi\|^2), \quad (8)$$

where $p_i(x) = \partial_\xi f(x; \phi)\big|_{\phi=\phi_0} = \left(\partial_\phi f(x; \phi)\big|_{\phi=\phi_0}\right) W_{:i}$, and $W_{:i}$ denotes the $i$-th column of $W$. The induced first-order linear modulation space of $\delta_\xi$ is therefore

$$\mathcal{P}_W := \text{span}\{p_i\}_{i=1}^{d_\xi}, \qquad \dim(\mathcal{P}_W) \leq \text{rank}(W) \leq d_\xi,$$

which is controlled by the context dimensionality $d_\xi$.

### 3.2. Identifiability Problem of Naive Correction

Despite the apparent generality, this mismatch correction suffers from a fundamental identifiability issue. Crucially, the objective in (5) is not well-defined in a pointwise sense, but only meaningful up to a smooth conjugacy class. Even in an idealized regime, there exist infinitely many pairs $(\Phi, \delta)$ that satisfy the conjugacy relation: indeed, for any auxiliary diffeomorphism $\Psi : \mathcal{X} \to \mathcal{X}$, one may define a reparameterized map $\tilde{\Phi} := \Phi \circ \Psi^{-1}$ together with a modified correction $\tilde{\delta} := \Psi_*(f_0 + \delta) - f_0$, yielding

$$\tilde{\Phi}_*(f_0 + \tilde{\delta}) = (\Phi \circ \Psi^{-1})_*(\Psi_*(f_0 + \delta)) = \Phi_*(f_0 + \delta) = g.$$

Thus, (5) admits an entire family of solutions connected by diffeomorphic reparameterization, often referred to as a smooth conjugacy orbit. As a consequence, uniqueness cannot be expected without an additional selection principle.

This ambiguity becomes even more severe in the data-driven setting. Because both the conjugacy map $\Phi_\theta$ and the correction term $\delta_\xi$ are learned jointly, there typically exist infinitely many parameter pairs $(\theta, \xi)$ that achieve comparably small training losses in (6), yet fail to realize the intended conjugacy relation (5) beyond the training set. Specifically, when the context dimension $d_\xi$ exceeds the codimension required to reconcile the source and target dynamics, the correction term becomes expressive enough to absorb not only the desired topology-altering discrepancy, but also essentially arbitrary variations, including those of diffeomorphic origin. A transparent manifestation of this pathology is the degenerate solution $\Phi_\theta \approx \text{Id}$, in which case the conjugacy objective in (6) collapses to standard trajectory matching, thereby forfeiting the structural guarantees and generalization benefits afforded by conjugacy-based modeling.

### 3.3. Why Orbit-Transverse Decomposition?

**Orthogonality as a minimality principle.** Jointly learning a corrective vector field together with a conjugating diffeomorphism is essentially non-identifiable, especially under finite observations. Resolving this ambiguity requires a

principled selection rule within the modulation space $\mathcal{P}_W$ in (8). The desired choice should be minimal yet complete: sufficiently expressive to capture the required topology-altering perturbations, while excluding redundant variations that can be represented by a diffeomorphic counterpart.

We enforce such minimality by encouraging *orthogonality* between $\mathcal{P}_W$ and the diffeomorphic directions induced by a family of parameterized normalizing flows. This promotes a structured decomposition of admissible dynamical variations into two conceptually distinct components: (i) *orbit* components, which move within a smooth conjugacy class of the reference vector field $f_0$ and therefore preserve its qualitative topology, and (ii) *transverse* components, which are necessarily transverse to this orbit and correspond to intrinsic, topology-changing modifications of $f_0$. These notions will be made precise in the following section.

To formalize how such an orthogonal decomposition can be imposed between these inherently nonlinear variations, we introduce the notion of the infinitesimal action of diffeomorphisms on $f_0$. This action provides a first-order linearization of diffeomorphic variations, yielding an orbit-tangent space at $f_0$. By equipping the space of vector fields with a chosen inner product, we can then characterize orthogonality to this tangent space. The resulting orthogonal complement serves as a principled proxy for directions that cannot be explained by infinitesimal diffeomorphisms, and therefore represent intrinsic, topology-altering perturbations of $f_0$.

**Definition 3.1** (Tangent space to the conjugacy orbit). Let $U \subset \mathbb{R}^{d_x}$ be an open set and let $\mathfrak{X}(U)$ be the space of vector fields on $U$. Let $\mathrm{Diff}(U)$ be the group of diffeomorphisms $\Phi : U \to U$, which acts on $f \in \mathfrak{X}(U)$ by pushforward:

$$(\Phi_* f)(y) = D\Phi(x)f(x), \qquad x = \Phi^{-1}(y).$$

Define the smooth conjugacy orbit of $f$:

$$\mathcal{O}_f := \{\Phi_* f : \Phi \in \mathrm{Diff}(U)\}.$$

The orbit-tangent directions at $f$ are the derivatives at $\varepsilon = 0$ of $(\Phi_\varepsilon)_* f$ along smooth curves $\{\Phi_\varepsilon\}_{|\varepsilon| \ll 1} \subset \mathrm{Diff}(U)$ with $\Phi_0 = \mathrm{Id}$. Then, the tangent space to the orbit is defined as

$$T_f \mathcal{O}_f := \left\{ \left.\frac{\mathrm{d}}{\mathrm{d}\varepsilon}\right|_{\varepsilon=0} (\Phi_\varepsilon)_* f : \{\Phi_\varepsilon\} \subset \mathrm{Diff}(U),\ \Phi_0 = \mathrm{Id} \right\}.$$

**Lemma 3.2.** *For any such curve $\{\Phi_\varepsilon\}$, define its generator:*

$$v(x) := \left.\frac{\mathrm{d}}{\mathrm{d}\varepsilon}\right|_{\varepsilon=0} \Phi_\varepsilon(x).$$

*Then,*

$$\left.\frac{\mathrm{d}}{\mathrm{d}\varepsilon}\right|_{\varepsilon=0} (\Phi_\varepsilon)_* f = [f, v],$$

*where $[f, v] = (Dv)f - (Df)v$ is the Lie bracket of vector fields. Hence, $T_f \mathcal{O}_f = \{[f, v] : v \in \mathfrak{X}(U)\}$.*

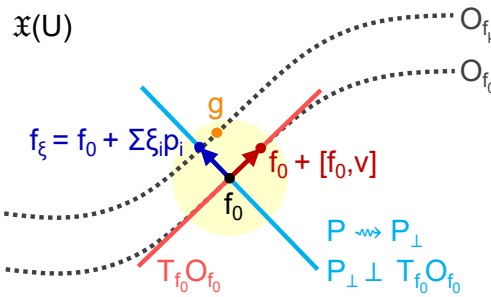

*Figure 1.* **Conceptual illustration of the proposed approach.** A target vector field $g$ may not be smoothly conjugate to a given reference $f_0$. Our approach therefore seeks a minimal correction $f_\xi \in f_0 + \mathcal{P}$ such that $g$ becomes smoothly conjugate to $f_\xi$. The dotted gray curve depicts the conjugacy orbit $\mathcal{O}_{f_0}$ of $f_0$, while the light-red line denotes the corresponding orbit-tangent space $T_{f_0} \mathcal{O}_{f_0}$ at $f_0$. The light-blue line represents a transverse modulation space $\mathcal{P}_\perp$, chosen to be orthogonal to $T_{f_0} \mathcal{O}_{f_0}$ under a specified inner product. This choice provides a practical representative of intrinsic, topology-altering variations modulo orbit-tangent directions. In practice, we enforce this transversality through a min-max regularizer that drives $\mathcal{P}$ toward such an orthogonal complement, i.e., $\mathcal{P} \rightsquigarrow \mathcal{P}_\perp \perp T_{f_0} \mathcal{O}_{f_0}$. The resulting correction yields $f_\xi$, whose conjugacy orbit $\mathcal{O}_{f_\xi}$ contains the target vector field $g$.

*Proof.* By the definition, $(\Phi_\varepsilon)_* f(y) = D\Phi_\varepsilon(x)f(x)$ with $x = \Phi_\varepsilon^{-1}(y)$. Differentiating at $\varepsilon = 0$ yields

$$\left.\frac{\mathrm{d}}{\mathrm{d}\varepsilon}\right|_{\varepsilon=0} (\Phi_\varepsilon)_* f = (Dv)f + Df\left(\left.\frac{\mathrm{d}}{\mathrm{d}\varepsilon}\right|_{\varepsilon=0} x\right),$$

and since $\left.\frac{\mathrm{d}}{\mathrm{d}\varepsilon}\right|_{\varepsilon=0} \Phi_\varepsilon^{-1} = -v$, the claim follows. $\square$

*Remark* 3.3. The conjugacy orbit $\mathcal{O}_f$ consists of all vector fields obtained by pushing forward $f$ under diffeomorphisms: motion along $\mathcal{O}_f$ therefore generates vector fields that remain smoothly conjugate to $f$, hence preserving all topological properties of $f$. The tangent space $T_f \mathcal{O}_f$ comprises the infinitesimal directions along this orbit, corresponding to first-order linearized variations of $f$ induced by perturbing the conjugating diffeomorphism away from the identity. Concretely, these directions are generated by Lie brackets $[f, v]$, where $v$ ranges over vector fields acting as infinitesimal generators of diffeomorphisms.

*Remark* 3.4. By Lemma 3.2, expanding in $\varepsilon$ yields

$$(\Phi_\varepsilon)_* f = (\Phi_0)_* f + \varepsilon \left.\frac{\mathrm{d}}{\mathrm{d}\varepsilon}\right|_{\varepsilon=0} (\Phi_\varepsilon)_* f + O(\varepsilon^2)$$

$$= f + \varepsilon[f, v] + O(\varepsilon^2).$$

From Remark 3.4, the context-modulated NODE in (8) admits the first-order decomposition of

$$(\Phi_\varepsilon)_* f_\xi = f_0 + \sum_{i=1}^{d_\xi} \xi_i p_i + \varepsilon[f_0, v] + O(\|\xi\|^2 + \varepsilon^2 + \varepsilon\|\xi\|).$$

Here, $\sum_{i=1}^{d_\xi} \xi_i p_i \in \mathcal{P}_W$ denotes the first-order modulation defined in (8), and $[f_0, v] \in T_{f_0} \mathcal{O}_{f_0}$. As noted earlier, in

the absence of any structural constraint, the modulation space $\mathcal{P}_W$ may be arbitrary and may contain redundant variations that can be represented by elements of the orbit-tangent $T_{f_0}\mathcal{O}_{f_0}$. Therefore, we define our identifiability principle by learning a space $\mathcal{P}_W \rightsquigarrow \mathcal{P}_\perp \perp T_{f_0}\mathcal{O}_{f_0}$ that is (approximately) orthogonal, under a chosen inner product (see Fig. 1). A standard choice is to equip the space of vector fields with a population $L^2$ inner product:

$$\langle u, v \rangle_\rho := \mathbb{E}_\rho\big[u(x)^\top v(x)\big] \qquad u, v \in \mathcal{H},$$

where $\mathcal{H} \subset L^2(\rho; \mathbb{R}^{d_x})$ denotes a Hilbert space of vector fields and $\rho$ is a chosen measure over the state space. Under this metric, orthogonality between modulation and orbit-tangent directions is expressed as

$$\langle p_i, [f_0, v] \rangle_\rho = 0 \qquad \forall p_i \in \mathcal{P}_W, \ [f_0, v] \in T_{f_0}\mathcal{O}_{f_0}. \quad (9)$$

In practice, we enforce a computable relaxation of this condition via projection-based regularization and an adversarial probing scheme, as described in the subsequent section.

**Why first-order orthogonality is sufficient.** The proposed selection rule raises a natural question: does enforcing first-order orthogonality, via restricting $\mathcal{P}_W$ to an orthogonal complement, retain sufficient local expressivity to restore smooth conjugacy to an arbitrary nearby target vector field $g$, which need not itself be smoothly conjugate to $f_0$?

*Unfolding theory* (Arnold, 2012) provides a standard answer. Under a finite-codimension assumption at an equilibrium, one can construct a finite-dimensional linear space of *unfoldings* that forms a transversal slice to the orbit-tangent space $T_{f_0}\mathcal{O}_{f_0}$. Moving along this unfolding space parameterizes the intrinsic degrees of freedom required to locally recover smooth conjugacy to $g$. Since orthogonality implies transversality under a non-degenerate inner product, our construction in (9) can be interpreted as a practical approximation of this unfolding basis (see Appendix A for a detailed discussion).

A second, equally important question then arises. The orbit-tangent space $T_{f_0}\mathcal{O}_{f_0}$ is infinite-dimensional and defined via the full diffeomorphism group, and is therefore not directly computable in practice. In the sequel, we address this issue by replacing the full orbit-tangent space with a model-induced, adversarially probed surrogate that can be efficiently estimated during training.

### 3.4. Adversarial Orthogonality Regularization

**Model-induced orbit-tangent subspace.** The full orbit-tangent space $T_{f_0}\mathcal{O}_{f_0}$, defined by the action of the *entire* diffeomorphism group, is *infinite-dimensional* therefore intractable. Instead, we impose orthogonality relative to a family of *model-induced* orbit-tangent subspace determined

by a chosen normalizing-flow architecture $\mathcal{A}_\Phi : \gamma \mapsto \Phi_\gamma$. Here, $\gamma \in \Gamma$ denotes an argument parameter indexing the architecture and is not required to coincide with $\theta$ used for learning the conjugating diffeomorphism in (6). The image $\mathcal{A}_\Phi(\Gamma) \subset \mathrm{Diff}(\mathcal{X})$ thus represents a class of diffeomorphisms *realizable* by the selected architecture.

We next introduce the infinitesimal generators associated with this realizable diffeomorphism family, and define a set of corresponding orbit-tangent directions induced by this family. Enforcing orthogonality to this model-induced orbit-tangent family removes precisely those vector-field variations that can be explained by diffeomorphic reparameterizations expressible within $\mathcal{A}_\Phi$, which is sufficient to resolve identifiability relative to the chosen model class.

Motivated by the layered structure of modern normalizing flows, we write $\Phi_\gamma = \Phi_\gamma^{(K)} \circ \cdots \circ \Phi_\gamma^{(1)}$, and introduce, for each layer, an auxiliary $\varepsilon$-parameterization path:

$$\Phi_{\gamma,\varepsilon}^{(k)} : \mathcal{X} \to \mathcal{X}, \qquad \Phi_{\gamma,0}^{(k)} = \mathrm{Id}, \quad \Phi_{\gamma,1}^{(k)} = \Phi_\gamma^{(k)},$$

which interpolates between the identity and the learned transformation. The corresponding layerwise infinitesimal generator is then defined by

$$v_\gamma^{(k)}(x) := \frac{\mathrm{d}}{\mathrm{d}\varepsilon}\bigg|_{\varepsilon=0} \Phi_{\gamma,\varepsilon}^{(k)}(x) \in \mathfrak{X}(\mathcal{X}).$$

For affine coupling layers (Dinh et al., 2017), this construction admits a particularly simple closed-form expression in terms of the scales and translations (see Appendix B). Letting $\varepsilon = (\varepsilon_1, \ldots, \varepsilon_K)$ and $\Phi_{\gamma,\varepsilon} := \Phi_{\gamma,\varepsilon_K}^{(K)} \circ \cdots \circ \Phi_{\gamma,\varepsilon_1}^{(1)}$, a first-order expansion around $\varepsilon = 0$ yields

$$\Phi_{\gamma,\varepsilon}(x) = x + \sum_{k=1}^{K} \varepsilon_k v_\gamma^{(k)}(x) + O\big(\|\varepsilon\|^2\big).$$

Thus, up to higher-order terms, the $\Phi_\gamma$-induced linear subspace of infinitesimal diffeomorphism is $\mathrm{span}\{v_\gamma^{(k)}\}$. Then, we define the orbit-tangent subspace at $f_0$ induced by $\Phi_\gamma$ as

$$T_{f_0}\mathcal{O}_{f_0}(\gamma) := \mathrm{span}\Big\{\big[f_0, v_\gamma^{(k)}\big]\Big\}_{k=1}^{K},$$

which represents realizable orbit-tangent directions for the chosen $\mathcal{A}_\Phi$ at parameter $\gamma$.[2]

**Worst-case realizable orbit-tangent direction.** Varying $\gamma$ induces a family of tangent subspaces $\{T_{f_0}\mathcal{O}_{f_0}(\gamma)\}_{\gamma \in \Gamma}$, each capturing the first order diffeomorphic variations that are realizable by the chosen $\mathcal{A}_\Phi$ at parameter $\gamma$. Accordingly, we need to enforce orthogonality of the modulation

---

[2]When using a continuous normalizing flow (Chen et al., 2018), the neural vector field $v_\gamma$ itself serves as an infinitesimal generator. The family $\{v_\gamma : \gamma \in \Gamma\}$ induces the realizable orbit-tangent set.

space $\mathcal{P}_W$ against this entire family, rather than against a slice generated by a single fixed $\gamma$. Formally, we define the set of realizable orbit-tangent directions as

$$\mathcal{R}_{\mathcal{A}_\Phi} := \left\{ r_{\gamma,\alpha} = \sum_{k=1}^K \alpha_k [f_0, v_\gamma^{(k)}] : \gamma \in \Gamma, \alpha \in \mathbb{R}^K \right\}.$$

This constructs an objective that regularizes $\mathcal{P}_W$ to be orthogonal to the *worst-case realizable direction* within $\mathcal{R}_{\mathcal{A}_\Phi}$, i.e., the direction most aligned with $\mathcal{P}_W$.

Formally, let $\Pi_{\mathcal{P}_W}$ denote the orthogonal projection onto $\mathcal{P}_W$. Exact $\mathcal{P}_W \perp \mathcal{R}_{\mathcal{A}_\Phi}$ means that, for every $r \in \mathcal{R}_{\mathcal{A}_\Phi}$,

$$\langle p, r \rangle = 0 \ \ \forall p \in \mathcal{P}_W \quad \Longleftrightarrow \quad \Pi_{\mathcal{P}_W} r = 0,$$

which recovers the desired condition in (9) for the restricted, model-induced orbit-tangent directions. This motivates the following worst-case alignment functional:

$$\mathcal{L}_{\text{orth}}(\mathcal{P}_W, \mathcal{R}_{\mathcal{A}_\Phi}) := \sup_{r_{\gamma,\alpha} \in \mathcal{R}_{\mathcal{A}_\Phi} \setminus \{0\}} \frac{\|\Pi_{\mathcal{P}_W} r_{\gamma,\alpha}\|^2}{\|r_{\gamma,\alpha}\|^2}. \quad (10)$$

This quantity $\mathcal{L}_{\text{orth}}$ is scale-invariant and equals $\sup_{r \in \mathcal{R}_{\mathcal{A}_\Phi}} \cos^2(\angle(r, \mathcal{P}_W))$, i.e., the squared cosine of the smallest principal angle between $\mathcal{P}_W$ and the realizable worst orbit-tangent direction. Consequently,

$$\mathcal{L}_{\text{orth}}(\mathcal{P}_W, \mathcal{R}_{\mathcal{A}_\Phi}) = 0 \quad \Longleftrightarrow \quad \mathcal{P}_W \perp \mathcal{R}_{\mathcal{A}_\Phi}.$$

Minimizing $\mathcal{L}_{\text{orth}}$ therefore enforces a strong minimality principle: any component of $\mathcal{P}_W$ that can be explained by a realizable diffeomorphism is penalized in the worst case.[3]

**Adversarial orthogonality.** To operationalize (10), we must approximate the supremum over admissible orbit-tangent directions, which is not available in closed form. We therefore introduce a min-max training scheme in the spirit of Generative Adversarial Networks (GANs) (Goodfellow et al., 2020) and adversarial regularization approaches (Yu et al., 2019). Combining with the conjugacy loss in (6), we define the following min-max objective:

$$\min_{\theta, W, \xi} \max_{\gamma, \alpha} \left[ \mathcal{L}_{\text{conj}}^{\text{flow}}(\theta, W, \xi) + \frac{\|\Pi_{\mathcal{P}_W}[f_0, v_{\gamma,\alpha}]\|^2}{\|[f_0, v_{\gamma,\alpha}]\|^2} \right], \quad (11)$$

where $v_{\gamma,\alpha} := \sum_{k=1}^K \alpha_k v_\gamma^{(k)}$ (thus $r_{\gamma,\alpha} = [f_0, v_{\gamma,\alpha}]$) is induced by a learnable normalizing flow $\Phi_\gamma$, while $\alpha \in \mathbb{R}^K$ forms a learnable mixture over these candidates. It is noteworthy that the role of $\Phi_\gamma$ is solely to adversarially scan for worst-case directions that maximize the alignment functional; it is distinct from the learnable diffeomorphism $\Phi_\theta$, which is trained to find the smooth conjugacy map.

---

[3]Appendix D also presents a numerically more stable *worst-case trace-ratio* alignment functional as an alternative surrogate objective, which we use in practice.

We evaluate the orthogonal regularizer on a data batch $\{x_b\}_{b=1}^B$ by stacking the modulation and orbit-tangent directions as $\tilde{P}_W = [P_W(x_1), \ldots P_W(x_B)]^\top \in \mathbb{R}^{Bd_x \times d_\xi}$, where $P_W(x_b) = [p_1(x_b), \ldots, p_{d_\xi}(x_b)] \in \mathbb{R}^{d_x \times d_\xi}$, and $\tilde{r}_{\gamma,\alpha} = [[f_0, v_{\gamma,\alpha}](x_1), \ldots, [f_0, v_{\gamma,\alpha}](x_B)]^\top \in \mathbb{R}^{Bd_x}$. We then estimate the continuous-space alignment functional as

$$\widehat{\mathcal{L}}_{\text{orth}}(W, \gamma, \alpha) = \frac{\|\Pi_{\mathcal{P}_W}[f_0, v_{\gamma,\alpha}]\|^2}{\|[f_0, v_{\gamma,\alpha}]\|^2} \approx \frac{\|\Pi_{\tilde{P}_W} \tilde{r}_{\gamma,\alpha}\|^2}{\|\tilde{r}_{\gamma,\alpha}\|^2}, \quad (12)$$

where $\Pi_{\tilde{P}_W}$ is the ridge-regularized projection operator, i.e., $\Pi_{\tilde{P}_W} = \tilde{P}_W(\tilde{P}_W^\top \tilde{P}_W + \lambda I)^{-1} \tilde{P}_W^\top$ with a small $\lambda > 0$.

# 4. Experiments

## 4.1. Common Setting.

**Basic protocol.** We first train a common source NODE model using a rich collection of trajectories sampled from the source dynamical system, ensuring that the learned NODE accurately captures the long-term topological structure. In contrast, the target data are provided in a realistic setting: only a small number of noisy observed trajectories from the target system are available. Each method is then trained to model the target dynamics under this limited supervision, using the common source model for initialization.

**Competitors.** We compare six conjugacy-based learning methods built on a common source model $f_0(\cdot) = f(\cdot; \phi_0)$:

*(i) Baseline conjugacy:* learn a diffeomorphism $\Phi_\theta$ that conjugates the fixed source dynamics $f_0$ to the target (Friedman et al., 2025; Sagodi & Park, 2025); *(ii) Direct Tuning (DT):* jointly optimize $(\phi, \theta)$ under the same conjugacy loss, starting from $\phi \leftarrow \phi_0$; *(iii) $L^2$-SP:* Same as direct tuning, but add an $L^2$-SP penalty (Xuhong et al., 2018) to keep the tuned source weight close to $\phi_0$; *(iv) Augmentation (Aug.):* keep $f_0$ fixed and learn an augmented vector field $f_0 + \delta_\xi$ with a neural correction $\delta_\xi$ (Yin et al., 2021b) jointly with $\Phi_\theta$ using conjugacy; *(v) Context Modulation (CM):* keep $\phi_0$ fixed and adapt the source weight via a low-rank modulation $\phi_0 \mapsto \phi_0 + W\xi$ (Kirchmeyer et al., 2022) and jointly learn $(\theta, W, \xi)$ using conjugacy loss; *(vi) CM with Adversarial Orthogonality (CM-AO, ours):* add an adversarial orthogonality to CM-based method so that $\mathcal{P}_W$ avoids worst-case orbit-tangent directions (see Appendix C for details).

## 4.2. Hopf Bifurcation

**Data generation.** We consider a Hopf normal form $(\dot{x}_1, \dot{x}_2) = f_\mu^{(\text{gt})}(x_1, x_2)$ (Marsden & McCracken, 2012):

$$f_\mu^{(\text{gt})}(x_1, x_2) = ((\mu - r^2)x_1 - x_2, x_1 + (\mu - r^2)x_2),$$

where $r^2 = x_1^2 + x_2^2$. We set $\mu_s = -0.1$ for the source system and $\mu_t = 0.1$ for the target system. Crossing $\mu = 0$

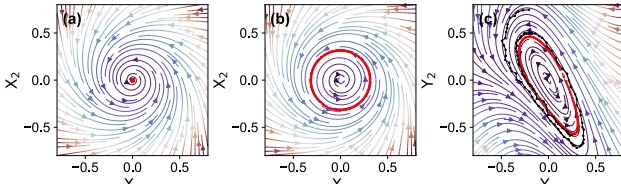

*Figure 2.* **Phase portraits of a Hopf bifurcating system**. Line colors indicate relative vector-field magnitude, and the converged invariant set is highlighted in red. (a) Source normal-form dynamics with $\mu_s = -0.1$. (b) Target normal-form dynamics with $\mu_t = 0.1$. (c) Target dynamics in observation space. A black curve indicates an example of observed trajectories.

*Table 1.* Invariant-set metrics for the Hopf problem in Section 4.2. Lower is better. Repeated five times.

| Method | Normal form $f_{\mu_t}$ | | | Observed $g = \Phi_* f_{\mu_t}$ | | |
|---|---|---|---|---|---|---|
| | $d_H$ | $W_2$ | MMD | $d_H$ | $W_2$ | MMD |
| Baseline | 0.317 | 0.316 | 0.757 | 0.547 | 0.381 | 0.851 |
| DT | 0.226 | 0.211 | 0.306 | 0.067 | 0.197 | 0.257 |
| $L^2$-SP | 0.229 | 0.214 | 0.314 | 0.068 | 0.189 | 0.249 |
| Aug. | 0.107 | 0.100 | 0.175 | 0.073 | 0.124 | 0.198 |
| CM | 0.142 | 0.119 | 0.177 | **0.064** | 0.131 | **0.173** |
| CM-AO | **0.081** | **0.071** | **0.109** | **0.064** | **0.121** | 0.182 |

induces a supercritical Hopf bifurcation: the origin changes from a stable focus to an unstable focus and a stable limit cycle emerges. The source dataset consisted of 64 noise-free trajectories integrated from the Hopf normal form with $\mu_s = -0.1$. For the target dataset, we first integrated trajectories $X(t)$ from the normal form with $\mu_t = 0.1$, then applied a diffeomorphism $\Phi^{(gt)}$ as $Y(t) = \Phi^{(gt)}(X(t)) + \eta(t)$, where $\eta(t)$ is Gaussian noise. This dataset is equal to integrating trajectories under the observational dynamics $g_{gt} = \Phi^{(gt)}_* f^{(gt)}_{\mu_t}$ with noise. We implemented $\Phi^{(gt)}$ as a nonlinear warping map. The target dataset contained only a single trajectory yielded by this procedure. See Appendix E.1 for details. Phase portraits are shown in Fig. 2.

**Results.** We first trained a common source NODE on the noiseless source dataset, obtaining a surrogate for the source dynamics shown in Fig. 2(a). We then trained each target model using the noisy target trajectories; a representative example is shown in Fig. 2(c). Training details and hyperparameter settings are provided in Appendix E.1.

After fitting each model to the target observation data, we numerically integrate trajectories from randomly sampled initial conditions over a sufficiently long horizon to obtain the corresponding invariant sets. Table 1 reports the Hausdorff distance ($d_H$), the 2-Wasserstein distance ($W_2$), and the Maximum Mean Discrepancy (MMD) between each predicted invariant set and the ground-truth one (see Appendix E.1 for the target-only baseline; see Appendix G for ablation studies). Overall, the proposed CM-AO achieves the best average performance on both the observed target dynamics

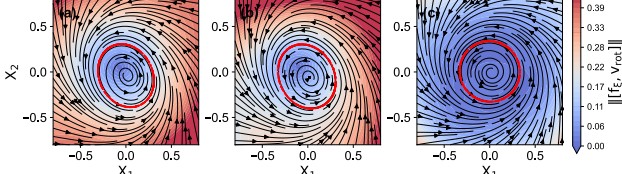

*Figure 3.* **Phase portraits of the learned corrected dynamics for a Hopf bifurcating system.** (a) Augmentation. (b) CM. (c) CM-AO. The background contour visualizes the Lie-bracket residual $\|[f_\xi, v_{rot}](x)\|_2$, quantifying the violation of SO(2) symmetry.

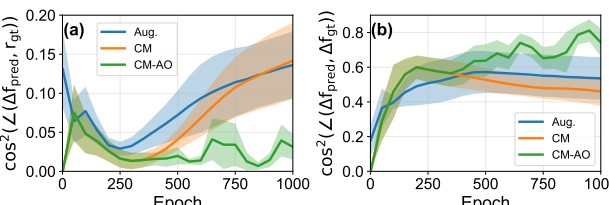

*Figure 4.* **Evolution of alignment metrics during training.** Squared cosine similarity between the learned correction and (a) the ground-truth orbit-tangent direction and (b) the intrinsic topology-changing direction. For CM-AO (green), orthogonality regularization is activated after a 300-epoch warm-up period.

$g = \Phi_* f_{\mu_t}$ and its normal form $f_{\mu_t}$. Notably, while the improvement in the observational space is marginal relative to augmentation and CM-based methods, the proposed method exhibits a substantially larger performance gap in the normal-form evaluation. This pattern suggests that the orthogonality constraint steers the learned source correction $f_0 + \mathcal{P}_W$ toward a minimal unfolding and prevents allocating correction capacity to diffeomorphic variations, whereas competing methods fail to isolate this minimal correction.

Fig. 3 compares the phase portraits of the corrected source dynamics $f_\xi$ learned by augmentation, CM, and CM-AO (see Appendix F.1 for other cases). The background contour shows the Lie bracket residual $\|[f_\xi, v_{rot}](x)\|_2$, where $v_{rot} = Jx$ is the generator of the SO(2) rotation. An ideal correction should recover the SO(2)-symmetric target normal form in Fig. 2(b), for which this residual vanishes. In augmentation and CM, the learned vector fields and the resulting limit cycles remain visibly deformed, indicating that the corrections retain diffeomorphic components. In contrast, CM-AO recovers a nearly SO(2)-symmetric vector field and a circular limit cycle close to the ground truth.

We further formalize this visual intuition by measuring the squared cosine similarity between the learned correction $\Delta f_{pred} = f_\xi - f_0$ and (i) the ground-truth orbit-tangent $r_{gt} = [f_0, v_\Phi^{(gt)}]$, as well as (ii) the intrinsic ground-truth bifurcation $\Delta f_{gt} = f^{(gt)}_{\mu_s} - f^{(gt)}_{\mu_t}$. In the ideal case, the former should converge to zero (perfect orthogonality), while the latter should be one (perfect alignment). As shown in Fig. 4, the orthogonality-regularized model exhibits this desired behavior consistently across training, whereas competing methods fail to achieve such a clear decomposition.

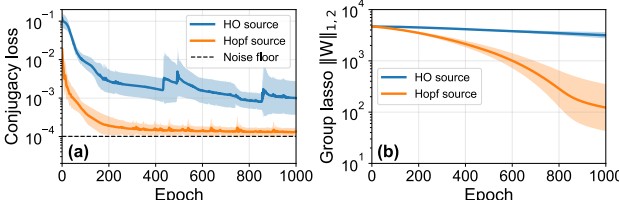

*Figure 5.* **Source-mismatch diagnostics.** Training dynamics on the Hopf benchmark using a structurally mismatched harmonic-oscillator (HO) source and a matched Hopf source. (a) Conjugacy loss and (b) group-lasso norm $\|W\|_{1,2}$ of the modulation weights.

**Mismatched source diagnosis.** The training dynamics of CM-AO also provide a diagnostic for source–target mismatch. In the unfolding view, a suitable source should represent the target as $g \approx (\Phi_\theta)_*(f_0 + \sum_i \xi_i p_i)$ through a diffeomorphism $\Phi_\theta$ and a small intrinsic correction $p_i \in \mathcal{P}_W$. Thus, the conjugacy loss measures reconstructability up to diffeomorphism, while the norm or sparsity of $W$ measures the complexity of the required correction. If both quantities fail to decrease, the target cannot be captured by a small unfolding around the chosen source $f_0$. We test this on the Hopf benchmark by using a harmonic oscillator as a mismatched source. As shown in Fig. 5, both the conjugacy loss and the group-lasso norm of $W$ plateau for the mismatched source, whereas they decrease for the Hopf source. Their joint stagnation indicates that the source is not a suitable structural anchor for the target dynamics.

### 4.3. Duffing Oscillator

**Data generation.** We consider a Hamiltonian Duffing oscillator (Strogatz, 2024) as our next benchmark:

$$(\dot{x}_1, \dot{x}_2) = f_\mu^{(\mathrm{gt})}(x_1, x_2) = (x_2, \mu x_1 - x_1^3),$$

where $\mu_s = -0.1$ for the source system and $\mu_t = 0.1$ for the target system. Crossing $\mu = 0$ induces a pitchfork-type bifurcation in the associated potential landscape: the system transitions from a single-well regime to a double-well regime, where two center equilibria emerge and the origin becomes an unstable saddle point. The data generation procedure closely follows the setup in Section 4.2 (see Appendix E.2 for details). See Fig. 6 for phase portraits.

**Results.** As in the Hopf benchmark, we first trained a source NODE on source trajectories to obtain a surrogate source dynamics shown in Fig. 6(a). We then trained each target model on noisy target trajectories (see Appendix E.2).

A Duffing oscillator, expressed in its canonical normal-form coordinates, exhibits a $\mathbb{Z}_2$ symmetry: the vector field is equivariant under the sign flip, $f(x) = -f(-x)$, as shown in Fig. 6(a–b). In the observed coordinates, however, the observation diffeomorphism breaks this symmetry as in Fig. 6(c). Motivated by this setup, we evaluate whether

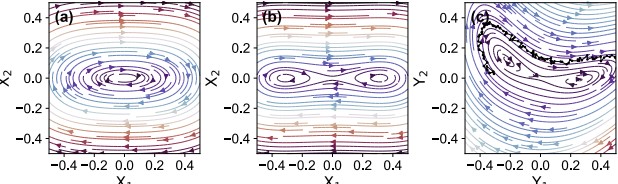

*Figure 6.* **Phase portraits of a Duffing oscillator.** (a) Source normal-form dynamics with $\mu_s = -0.1$. (b) Target normal-form dynamics with $\mu_t = 0.1$. (c) Target dynamics in observation space. A black curve indicates an example of observed trajectories.

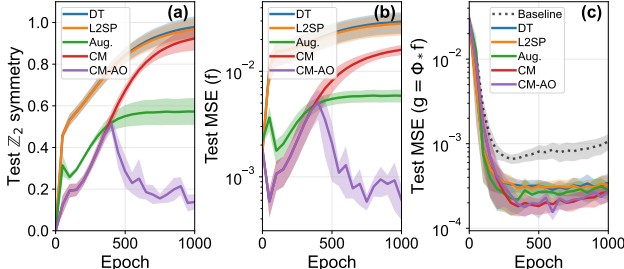

*Figure 7.* **Evolution of test symmetry and MSE metrics for learned vector fields.** (a) $\mathbb{Z}_2$ symmetry score. (b) Test MSE in the normal-form space. (c) Test MSE in the observation space. For CM-AO (purple), orthogonality regularization is activated after a 300-epoch warm-up period. Repeated five times.

the learned correction can recover the underlying $\mathbb{Z}_2$ symmetry when training from a symmetric source model but asymmetric, symmetry-broken observations. Concretely, after fitting each model, we compute a symmetry score as $\frac{\sqrt{\mathbb{E}\|f_{\mathrm{pred}}(x) + f_{\mathrm{pred}}(-x)\|^2}}{\sqrt{\mathbb{E}\|f_{\mathrm{pred}}(x)\|^2}}$. We additionally report the standard Mean Squared Error (MSE) between the modeled vector field and the ground-truth one also, since even a trivial estimate (e.g., a nearly zero field) can yield a vanishing symmetry score. We exclude the baseline conjugacy method from this comparison, since it leaves the source vector field unchanged. As shown in Fig. 7, the proposed CM-AO achieves the lowest test MSE and successfully uncovers the hidden $\mathbb{Z}_2$ symmetry of the observed target system (refer to Appendix F.2 for learned phase portrait examples).

CM-based methods naturally induce a low-dimensional, context-parameterized family of vector fields, $f_\xi(\cdot) = f(\cdot; \phi_0 + W\xi)$. This formulation allows us to assess whether variations of the learned vector field under context modulation are consistent with genuine topological changes in the underlying system. Concretely, let $\xi = \hat{\xi}$ explicitly denote the learned context vector that corrects the source dynamics. Since the uncorrected source corresponds to $\xi = 0$, the displacement $\hat{\xi} - 0$ defines a linear modulation direction within this $\xi$-parameterized family. Using this direction, we generate a one-parameter family of vector fields by sweeping $\xi$ as $\alpha\hat{\xi}$ with $\alpha \in [-1, 2]$. Then, for each $\alpha$, we compute the equilibria $\{x_0 : f_{\alpha\hat{\xi}}(x_0) = 0\}$ via the inverse optimization procedure of (Huh et al., 2025; Ren

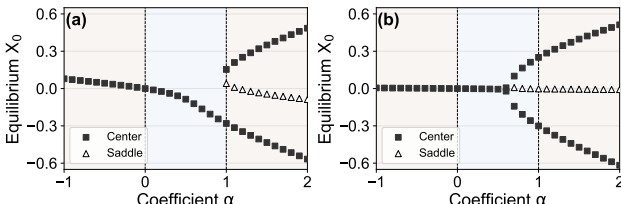

*Figure 8.* **Bifurcation diagrams identified from CM-based conjugacy models.** (a) Baseline Context Modulation (CM). (b) CM with adversarial orthogonality regularization (CM-AO).

*Table 2.* Invariant-set metrics for the Hopf extrapolation problem in Section 4.4. Lower is better. Repeated five times.

| | Normal form $f_{\mu_t}$ | | | Observed $g = \Phi_* f_{\mu_t}$ | | |
|---|---|---|---|---|---|---|
| Method | $d_H$ | $W_2$ | MMD | $d_H$ | $W_2$ | MMD |
| Baseline | 0.317 | 0.316 | 0.757 | 0.636 | 0.441 | 0.880 |
| DT | 0.487 | 0.392 | 0.494 | 0.485 | 0.461 | 0.523 |
| $L^2$-SP | 0.361 | 0.308 | 0.402 | 0.315 | 0.397 | 0.425 |
| Aug. | 0.347 | 0.313 | 0.789 | 0.666 | 0.394 | 0.851 |
| CM | 0.177 | 0.130 | 0.266 | 0.234 | **0.226** | 0.365 |
| CM-AO | **0.103** | **0.090** | **0.189** | **0.186** | **0.226** | 0.325 |

et al., 2020). Fig. 8 shows the resulting bifurcation diagrams, showing equilibrium locations and stability types as functions of $\alpha$ for (a) CM and (b) CM-AO. The results show that the proposed CM-AO recovers the correct symmetric pitchfork bifurcation, whereas the baseline CM exhibits a spurious, symmetry-broken branching pattern.

### 4.4. Hopf Bifurcation with Phase-Space Extrapolation

The previous benchmarks mainly assess *parameter-level* generalization, namely whether CM-AO can recover the bifurcation structure responsible for topological change. We further evaluate *phase-space* generalization. Specifically, we revisit the Hopf benchmark in Section 4.2 under an explicit initial-condition extrapolation setting, where target trajectories are sampled only from initial conditions far from the limit cycle and observed over a short time horizon. As a result, the training data do not directly expose the limit-cycle geometry. See Appendix E.3 for details.

Because the observed trajectories do not reach the relevant topological structure, applying the orthogonality regularization only on the observed data mainly constrains the outer phase-space region, rather than the topologically meaningful equilibrium structure. To address this, we introduce a simple self-regularization strategy. During training, we roll out the learned CM-AO dynamics $f_\xi$ over a longer horizon and use the resulting trajectories to augment the domain on which the regularization is applied (see Appendix E.3 for details). As shown in Table 2, CM-AO outperforms all competing methods, indicating that it recovers the underlying canonical structure rather than merely fitting short observed trajectories, thereby reconstructing the correct limit cycle from off-limit-cycle initializations.

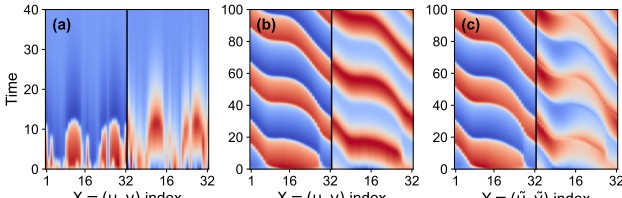

*Figure 9.* **FitzHugh–Nagumo system.** (a) Source dynamics ($\mu_s = 0$). (b) Target dynamics ($\mu_t = 0.6$). (c) Observed target dynamics.

*Table 3.* FitzHugh–Nagumo benchmark results in canonical $(u, v)$ coordinates. Repeated five times. See Appendix F.4 for details.

| | $\omega = 1$ (Interp.) | | | $\omega = 2$ (Extrap.) | | |
|---|---|---|---|---|---|---|
| Method | $d_H$ | $W_2$ | MSE | $d_H$ | $W_2$ | MSE |
| Baseline | 12.29 | 11.87 | 2.199 | 12.92 | 12.59 | 2.198 |
| DT | 160.0 | 123.3 | 199.0 | 29.70 | 12.46 | 6.080 |
| $L^2$-SP | 204.1 | 181.1 | 408.9 | 97.24 | 26.10 | 8.119 |
| Aug. | 9.717 | 9.202 | 2.218 | 9.725 | 6.114 | 1.513 |
| CM | 6.993 | 4.949 | 0.199 | 10.38 | 7.849 | 0.703 |
| CM-AO | **4.644** | **4.034** | **0.177** | **8.941** | **2.806** | 0.218 |

### 4.5. FitzHugh–Nagumo Reaction–Diffusion

**Data generation.** We consider the FitzHugh–Nagumo system (Rocsoreanu et al., 2012) discretized on a periodic grid, yielding a 64-dimensional ODE:

$$(\dot{u}_j, \dot{v}_j) = (D_u \Delta u_j + R(u_j) - v_j + \mu, \zeta(u_j + a - bv_j)),$$

for $j = 1, ..., 32$, where $R(u) = u - u^3/3$ and $(D_u, \zeta, a, b)$ are fixed. The source parameter $\mu_s = 0$ yields a homogeneous fixed-point regime (Fig. 9(a)), whereas the target parameter $\mu_t = 0.6$ produces traveling-wave spatiotemporal oscillations (Fig. 9(b)). We apply a nonlinear diffeomorphism to the target trajectories to obtain the observation-space dynamics (Fig. 9(c)). See Appendix E.4 for details.

**Results.** We first trained a source NODE on source trajectories, and then trained each target model on observed target trajectories initialized from phase-controlled sinusoidal profiles. Here, $\omega$ denotes the number of spatial phase cycles across the grid at the initial time. We used $\omega = 1$ for target training, corresponding to the interpolative setting, and held out $\omega = 2$ for phase-space extrapolation testing (see Appendix E.4). Table 3 shows that CM-AO outperforms all competitors, with the largest gains in the $\omega = 2$ case.

## 5. Conclusion

We address non-identifiability in conjugacy learning under topological mismatch between source and target dynamics. CM-AO combines low-rank modulation with adversarial orthogonality to quotient out orbit-tangent variations and recover minimal unfoldings. Across benchmarks, it reconstructs normal-form invariant sets and symmetries from distorted observations. Future work will extend this approach to higher-dimensional and nonlocal topological corrections.

## Acknowledgements

Changwook Jeong acknowledges support from the Samsung Research Funding & Incubation Center for Future Technology of Samsung Electronics (SRFC-IT2502-01); Samsung Electronics (IO250618-13097-01); the KIAT grant funded by MOTIE, Korea (P0023703, HRD Program for Industrial Innovation); the IITP grant funded by MSIT, Korea (RS-2020-II201336, AIGS); and the 2026 Research Fund of UNIST (Ulsan National Institute of Science & Technology) (1.260011.01).

## Impact Statement

This work contributes to machine learning for scientific modeling by addressing non-identifiability in conjugacy-based learning of dynamical systems. The proposed framework aims to improve the recovery of qualitative dynamical structures, including invariant sets, symmetries, and bifurcation behavior, from limited and distorted observations. As with other data-driven modeling approaches, its reliability depends on data quality, the suitability of the source model, and careful validation before use in safety-critical applications. We do not foresee immediate negative societal impacts beyond these general considerations.

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

# A. Versal Unfolding Theory

**Theorem A.1** (Versal unfolding, informal). *Let $x_0 \in \mathbb{R}^{d_x}$ and let $U$ be a sufficiently small neighborhood of $x_0$. Fix $f_0 \in \mathfrak{X}(U)$ with $f_0(x_0) = 0$, and consider the local action of*

$$\mathrm{Diff}_{x_0}(U) := \{\Phi \in \mathrm{Diff}(U) : \Phi(x_0) = x_0\}$$

*on $\mathfrak{X}(U)$ by pushforward. Then, its local conjugacy orbit is*

$$\mathcal{O}_{f_0} := \{\Phi_* f_0 : \Phi \in \mathrm{Diff}_{x_0}(U)\},$$

*and the corresponding tangent space is*

$$T_{f_0}\mathcal{O}_{f_0} := \{[f_0, v] : v \in \mathfrak{X}(U), v(x_0) = 0\}.$$

*Assume that $f_0$ has finite codimension $d_\mu$ under this local action, i.e., the quotient space $\mathfrak{X}(U)/T_{f_0}\mathcal{O}_{f_0}$ is $d_\mu$-dimensional. Choose representatives $u_1, \ldots, u_{d_\mu} \in \mathfrak{X}(U)$ whose equivalence classes $[u_i]$ form a basis of this quotient. Their span $\mathcal{S} := \mathrm{span}\{u_1, \ldots, u_{d_\mu}\}$ serves as a chosen transversal slice transverse to $T_{f_0}\mathcal{O}_{f_0}$, used to represent the quotient directions. The associated $\mu$-parameter unfolding*

$$f_\mu := f_0 + \sum_{i=1}^{d_\mu} \mu_i u_i, \qquad \mu \in \mathbb{R}^{d_\mu}$$

*is versal in the following sense: for any $g \in \mathfrak{X}(U)$ that is sufficiently close to $f_0$ in $C^k$ (for $k$ large enough), there exist parameters $\mu(g) \in \mathbb{R}^{d_\mu}$, a possibly smaller neighborhood $U' \subset U$ containing $x_0$, and a diffeomorphism $\Phi \in \mathrm{Diff}_{x_0}(U')$ such that*

$$g\big|_{U'} = \Phi_* f_{\mu(g)}.$$

*That is, $g$ is smoothly conjugate on a neighborhood of $x_0$ to some member of the versal family $\{f_\mu\}_{\mu \in \mathbb{R}^{d_\mu}}$.*

For a comprehensive treatment of Theorem A.1, we refer to the reader to standard texts in dynamical systems theory, including (Arnold, 2012; Golubitsky et al., 2012; Guckenheimer & Holmes, 2013; Kuznetsov, 1998).

Intuitively, the quotient space $\mathfrak{X}(U)/T_{f_0}\mathcal{O}_{f_0}$ captures intrinsic perturbations of the reference vector field $f_0$ that cannot be generated by infinitesimal diffeomorphic transformations along the orbit-tangent space $T_{f_0}\mathcal{O}_{f_0}$. As such, it admits a natural interpretation as the local space of *unfolding*: it parameterizes the degrees of freedom that genuinely alter the smooth conjugacy class of $f_0$ at first order. Specifically, Theorem A.1 guarantees the existence of a finite-dimensional *versal* family

$$f_\mu = f_0 + \sum_{i=1}^{d_\mu} \mu_i u_i \in f_0 + \mathcal{S},$$

such that any vector field $g$ sufficiently close to $f_0$, though not necessarily smoothly conjugate to $f_0$ itself, is nevertheless smoothly conjugate (locally near $x_0$) to some member of this family. Importantly, both the orbit-tangent space $T_{f_0}\mathcal{O}_{f_0}$ and the chosen unfolding directions $\mathcal{S} = \mathrm{span}\{u_1, \ldots, u_{d_\mu}\}$ are linear spaces defined at the level of first-order variations. This observation justifies why, for sufficiently small discrepancies, it is adequate in our framework to adjust and regularizes the source vector field only at the first-order level.

While Theorem A.1 guarantees the existence of a linear unfolding space that locally captures the intrinsic directions of variation, such a space is not uniquely determined: there remains freedom in the choice of a particular representative of the unfolding directions. The sole requirement is that the chosen spanning set $\mathcal{S}$ defines an appropriate transversal slice to the orbit-tangent $T_{f_0}\mathcal{O}_{f_0}$. When an inner product structure is available, a particularly natural choice is the orthogonal complement of $T_{f_0}\mathcal{O}_{f_0}$. This choice is minimal in the sense that it removes all infinitesimal variations attributable to diffeomorphic reparameterizations with respect to the chosen metric, while remaining complete in that it retains all directions necessary to represent intrinsic perturbations of the vector field modulo smooth conjugacy. This perspective provides the theoretical motivation underlying our proposed framework.

Motivated by the preceding discussion, we introduce the notion of identifiability used in this work.

**Definition A.2** (Local first-order identifiability). Let $\mathcal{H}$ be a Hilbert space of vector fields equipped with an inner product $\langle \cdot, \cdot \rangle$, and let $T_{f_0}\mathcal{O}_{f_0} \subset \mathcal{H}$ denote the orbit-tangent space of the smooth conjugacy orbit at $f_0$. A candidate modulation space $\mathcal{P} \subset \mathcal{H}$ is said to be locally identifiable to first order modulo diffeomorphic variation if

$$\mathcal{P} \cap T_{f_0}\mathcal{O}_{f_0} = \{0\}.$$

Equivalently, every admissible first-order discrepancy

$$h \in \mathcal{P} + T_{f_0}\mathcal{O}_{f_0}$$

admits a unique decomposition:

$$h = p + t, \qquad p \in \mathcal{P}, \quad t \in T_{f_0}\mathcal{O}_{f_0}.$$

In this case, the component $p \in \mathcal{P}$ is the identifiable representative of the intrinsic correction modulo infinitesimal diffeomorphic variation.

In our setting, $h$ denotes the first-order source–target discrepancy near $f_0$, so that

$$g \approx f_0 + h = f_0 + p + t, \qquad p \in \mathcal{P}, \quad t \in T_{f_0}\mathcal{O}_{f_0}.$$

For example, the first-order expansion in the main text corresponds to $p = \sum_i \xi_i p_i$ and $t = \varepsilon[f_0, v]$. The condition $P \cap T_{f_0}\mathcal{O}_{f_0} = \{0\}$ rules out any nonzero modulation direction that can also be realized as an infinitesimal diffeomorphic variation. Therefore, whenever the discrepancy admits such a first-order representation, the intrinsic component $p$ is uniquely identified modulo orbit-tangent directions. A sufficient way to impose this direct-sum condition is to choose $\mathcal{P}$ orthogonal to $T_{f_0}\mathcal{O}_{f_0}$ under the chosen metric:

$$\mathcal{P} \perp T_{f_0}\mathcal{O}_{f_0}.$$

This motivates the adversarial orthogonality regularization used in this paper.

*Remark* A.3 (Model-relative and empirical identifiability). The identifiability condition in Definition A.2 is an ideal first-order condition stated with respect to the full orbit-tangent space $T_{f_0}\mathcal{O}_{f_0}$ and the chosen inner product on $\mathcal{H}$. In practice, however, the proposed method enforces this condition only approximately. First, the full orbit-tangent space induced by the entire diffeomorphism group is replaced by a model-induced family of realizable orbit-tangent directions generated by the chosen normalizing flow class. Thus, the resulting notion of identifiability is relative to the expressive power of this diffeomorphism model. Second, we equip the space of vector fields with the population $L^2(\rho)$ inner product:

$$\langle u, v \rangle_\rho = \mathbb{E}_\rho\big[u(x)^\top v(x)\big], \qquad u, v \in \mathcal{H},$$

where $\mathcal{H} \subset L^2(\rho; \mathbb{R}^{d_x})$. However, during training, this inner product is approximated by the empirical inner product:

$$\langle u, v \rangle_{\tilde{\rho}} = \frac{1}{B} \sum_{b=1}^{B} u(x_b)^\top v(x_b),$$

computed on a finite batch of sampled states $\{x_b\}_{b=1}^{B}$. Thus, the orthogonality condition is enforced only on the sampled domain. The resulting decomposition should therefore be interpreted as model-relative, empirical first-order identifiability.

This interpretation has two practical implications. First, the adversarial diffeomorphism class should be expressive enough to probe the relevant orbit-tangent directions that could otherwise be absorbed by the modulation space. Second, the empirical inner product should be evaluated on states covering the topologically relevant regions of the phase space. Otherwise, the orthogonality constraint may certify identifiability only on an observationally restricted region, leaving diffeomorphic ambiguities unresolved near the invariant sets or bifurcation structures of interest. When the observed trajectories do not reach these regions, the regularization domain should be augmented using prior knowledge, additional sampling, or rollouts of the learned dynamics during training.

## B. Model-Induced Orbit-Tangent Directions and Their Construction

This appendix provides the technical details for approximating the orbit-tangent subspace induced by a learnable diffeomorphic normalizing flow, with particular focus on architectures based on affine coupling layers (Dinh et al., 2017).

**Layerwise $\varepsilon$-parameterization.** Consider a normalizing-flow diffeomorphism expressed as a composition of $K$ layers:

$$\Phi_\gamma = \Phi_\gamma^{(K)} \circ \cdots \circ \Phi_\gamma^{(1)},$$

where each $\Phi_\gamma^{(k)} : \mathcal{X} \to \mathcal{X}$ is itself a diffeomorphism. For each layer, we introduce an auxiliary $\varepsilon$-parameterization path:

$$\Phi_{\gamma,\varepsilon}^{(k)} : \mathcal{X} \to \mathcal{X}, \qquad \Phi_{\gamma,0}^{(k)} = \mathrm{Id}, \quad \Phi_{\gamma,1}^{(k)} = \Phi_\gamma^{(k)},$$

which interpolates between the identity ($\varepsilon = 0$) and the learned transformation ($\varepsilon = 1$). Different choices of such paths yield generators that coincide to first order, and therefore induce the same first-order orbit-tangent directions.

**Layerwise infinitesimal generators.** The associated infinitesimal generator associated is defined by

$$v_\gamma^{(k)}(x) := \left.\frac{\mathrm{d}}{\mathrm{d}\varepsilon}\right|_{\varepsilon=0} \Phi_{\gamma,\varepsilon}^{(k)}(x) \in \mathfrak{X}(\mathcal{X}).$$

By construction, $v_\gamma^{(k)}$ is a vector field that captures the first-order effect of activating the $k$-th diffeomorphic layer from the identity. That is, the first-order expansion around $\varepsilon = 0$ yields

$$\Phi_{\gamma,\varepsilon}^{(k)}(x) = x + \varepsilon v_\gamma^{(k)}(x) + O(|\varepsilon|^2).$$

**First-order spanning of the composed diffeomorphism.** Let $\varepsilon = (\varepsilon_1, \ldots, \varepsilon_K)$ and define the composed path:

$$\Phi_{\gamma,\varepsilon} := \Phi_{\gamma,\varepsilon_K}^{(K)} \circ \cdots \circ \Phi_{\gamma,\varepsilon_1}^{(1)}.$$

Expanding the above to first order around $\varepsilon = 0$ yields

$$\Phi_{\gamma,\varepsilon}(x) = x + \sum_{k=1}^{K} \varepsilon_k \, v_\gamma^{(k)}(x) + O(\|\varepsilon\|^2),$$

so that, up to higher-order terms, the model-induced infinitesimal directions at parameter $\gamma$ lie in

$$\mathcal{V}(\gamma) := \mathrm{span}\left\{v_\gamma^{(k)}\right\}_{k=1}^{K} \subset \mathfrak{X}(\mathcal{X}).$$

**Induced orbit-tangent subspace.** Given a reference vector field $f_0$, the infinitesimal action of a diffeomorphism generated by $v$ induces the Lie bracket $[f_0, v]$. Consequently, the model-induced orbit-tangent subspace associated with $\mathcal{V}(\gamma) = \mathrm{span}\{v_\gamma^{(k)}\}_{k=1}^{K}$ is given by

$$T_{f_0}\mathcal{O}_{f_0}(\gamma) := \mathrm{span}\left\{\left[f_0, v_\gamma^{(k)}\right]\right\}_{k=1}^{K} \subset T_{f_0}\mathcal{O}_{f_0},$$

which is at most $K$-dimensional and approximates the orbit-tangent directions realizable by the normalizing flow at parameter $\gamma$. The Lie bracket is evaluated using Jacobian–Vector Products (JVPs):

$$\left[f_0, v_\gamma^{(k)}\right](x) = Dv_\gamma^{(k)}(x)f_0(x) - Df_0(x)v_\gamma^{(k)}(x),$$

where both terms can be computed with automatic differentiation without explicitly forming Jacobian matrices.

**Example: affine coupling layers.** For affine coupling layers (Dinh et al., 2017), a core building block of Real Non-Volume Preserving (Real NVP) flows, the layerwise generators admit a closed-form expression, enabling an especially efficient construction of the orbit-tangent directions. Consider an affine coupling layer with a split state $x = (x_1, x_2)$:

$$\Phi_\gamma^{(k)}(x_1, x_2) = \left(x_1, x_2 \odot \exp s_\gamma(x_1) + t_\gamma(x_1)\right).$$

A convenient $\varepsilon$-parameterization is obtained by scaling the outputs of the scale and translation networks:

$$\Phi_{\gamma,\varepsilon}^{(k)}(x_1, x_2) = \left(y_1(\varepsilon), y_2(\varepsilon)\right) = \left(x_1, x_2 \odot \exp\left(\varepsilon s_\gamma^{(k)}(x_1)\right) + \varepsilon t_\gamma^{(k)}(x_1)\right),$$

which clearly satisfies $\Phi_{\gamma,0}^{(k)} = \mathrm{Id}$ and $\Phi_{\gamma,1}^{(k)} = \Phi_\gamma^{(k)}$. Differentiating at $\varepsilon = 0$ gives

$$\frac{\mathrm{d}}{\mathrm{d}\varepsilon}\bigg|_{\varepsilon=0} y_1(\varepsilon) = 0,$$

and

$$\frac{\mathrm{d}}{\mathrm{d}\varepsilon}\bigg|_{\varepsilon=0} y_2(\varepsilon) = x_2 \odot \frac{\mathrm{d}}{\mathrm{d}\varepsilon}\bigg|_{\varepsilon=0} \exp\left(\varepsilon s_\gamma^{(k)}(x_1)\right) + \frac{\mathrm{d}}{\mathrm{d}\varepsilon}\bigg|_{\varepsilon=0} \varepsilon t_\gamma^{(k)}(x_1)$$
$$= x_2 \odot s_\gamma(x_1) + t_\gamma(x_1),$$

so the corresponding generator admits the closed form of

$$v_\gamma^{(k)}(x) = \left(0, x_2 \odot s_\gamma^{(k)}(x_1) + t_\gamma^{(k)}(x_1)\right).$$

## C. Formulations and Objectives of Baselines and Variants

**Common source model.** Let $\{X_n\}_{n=1}^N$ be a collection of training trajectories from the source dynamical system, where $X_n = \{x_n^0, ..., x_n^T\}$. We train a source NODE vector field $f(\cdot; \phi)$ by minimizing a standard trajectory-matching objective:

$$\min_\phi \mathcal{L}_{\mathrm{traj}}^{\mathrm{flow}}(\phi), \qquad \mathcal{L}_{\mathrm{traj}}^{\mathrm{flow}} = \frac{1}{NT} \sum_{n=1}^N \sum_{t=1}^T \|x_n^t - \varphi_f^{t\Delta\tau}(x_n^0)\|^2.$$

where $\varphi_f^{t\Delta\tau}(x_n^0)$ denotes the flow induced by $f(\cdot, \phi)$, i.e., the state obtained by integrating $f(\cdot; \phi)$ from the initial condition $x_n^0$ during time $t\Delta\tau$. The optimized parameters, denoted by $\phi_0$, define the common source model $f_0(\cdot) = f(\cdot; \phi_0)$ used to initialize all subsequent target-side transfer learning baselines.

**Baseline conjugacy learning.** Let $\{Y_n\}_{n=1}^N$ be a collection of target trajectories, where $Y_n = \{y_n^0, ..., y_n^T\}$. A learnable diffeomorphism $\Phi_\theta$, parameterized by a normalizing flow (Papamakarios et al., 2021), is trained to push forward the source dynamics to the target via a flow-level smooth conjugacy loss. The objective consists of a conjugacy loss $\mathcal{L}_{\mathrm{conj}}^{\mathrm{flow}}$ together with a near-identity penalty $\Omega_\Phi(\theta)$ (Friedman et al., 2025; Sagodi & Park, 2025):

$$\min_\theta \left[ \mathcal{L}_{\mathrm{conj}}^{\mathrm{flow}}(\theta) + \Omega_\Phi(\theta) \right],$$

where

$$\mathcal{L}_{\mathrm{conj}}^{\mathrm{flow}} = \frac{1}{NT} \sum_{n=1}^N \sum_{t=1}^T \left\| y_n^t - \Phi_\theta(\varphi_{f_0}^{t\Delta\tau}(\Phi_\theta^{-1}(y_n^0))) \right\|^2, \qquad \Omega_\Phi = \lambda_\Phi \cdot \frac{1}{NT} \sum_{n=1}^N \sum_{t=1}^T \|\Phi_\theta^{-1}(y_n^t) - y_n^t\|^2.$$

The source parameters $\phi_0$ are kept fixed, and no explicit correction is applied to the source vector field for this method. During target-side training, only the diffeomorphism parameters $\theta$ are optimized.

**Conjugacy with Direct Tuning (DT).** This variant jointly updates the source parameters $\phi$ and the diffeomorphism parameters $\theta$, starting from $\phi \leftarrow \phi_0$, by minimizing the same conjugacy objective:

$$\min_{\theta,\phi} \left[ \mathcal{L}_{\mathrm{conj}}^{\mathrm{flow}}(\theta, \phi) + \Omega_\Phi(\theta) \right],$$

where $\mathcal{L}_{\mathrm{conj}}^{\mathrm{flow}}(\theta, \phi)$ denotes the smooth conjugacy loss evaluated using the trainable source dynamics $f(\cdot, \phi)$, in place of $f_0$.

**Conjugacy with $L^2$-SP.** $L^2$-SP augments conjugacy-based direct tuning with a parameter-anchoring regularizer (Xuhong et al., 2018) that penalizes deviation from the source parameters:

$$\min_{\theta,\phi} \left[ \mathcal{L}_{\mathrm{conj}}^{\mathrm{flow}}(\theta, \phi) + \Omega_\Phi(\theta) + \lambda_{\mathrm{SP}} \cdot \|\phi - \phi_0\|^2 \right],$$

**Conjugacy with augmentation (Aug.).** This method introduces a neural corrective field $\delta_\xi$, parameterized by a standard neural network, and jointly learns a diffeomorphism $\Phi_\theta$ using the augmented vector field $f_\xi = f_0 + \delta_\xi$:

$$\min_{\theta,\xi} \left[ \mathcal{L}_{\text{conj}}^{\text{flow}}(\theta,\xi) + \Omega_\Phi(\theta) + \Omega_\delta(\xi) \right],$$

where $\mathcal{L}_{\text{conj}}^{\text{flow}}(\theta,\xi)$ denotes the smooth conjugacy loss evaluated under the augmented model $f_\xi$. Following (Yin et al., 2021b), we regularize the correction term $\delta_\xi$ by an $L^2$ complexity penalty $\Omega_\delta$. Because $\delta_\xi$ is defined on the source state space $\mathcal{X}$, its $L^2$ regularization is applied in the pullback coordinates $x_n^t = \Phi_\theta^{-1}(y_n^t)$ (or, for $x \sim p(x)$ drawn from a predefined density on the source coordinates):

$$\Omega_\delta(\xi) = \lambda_\delta \cdot \frac{1}{NT} \sum_{n=1}^{N} \sum_{t=1}^{T} \|\delta_\xi(x_n^t)\|^2.$$

During training, $\phi_0$ remains fixed and only the diffeomorphism and correction parameters $(\theta, \xi)$ are optimized.

**Conjugacy with Context Modulation (CM).** Context modulation (Kirchmeyer et al., 2022) adapts the source model via a low-rank parameter update $\phi_0 \mapsto \phi_0 + W\xi$, yielding the modulated vector field $f_\xi(\cdot) = f(\cdot; \phi_0 + W\xi)$. We jointly learn the modulation $W\xi$ and a diffeomorphism $\Phi_\theta$ by optimizing

$$\min_{\theta,W,\xi} \left[ \mathcal{L}_{\text{conj}}^{\text{flow}}(\theta, W, \xi) + \Omega_\Phi(\theta) + \Omega_{W,\xi}(W, \xi) \right],$$

where $\mathcal{L}_{\text{conj}}^{\text{flow}}(\theta, W, \xi)$ is the smooth conjugacy loss evaluated using the modulated dynamics $f_\xi$. The regularizer $\Omega_{W,\xi}$ encourages sparse modulation through a group-Lasso penalty on $W$ and controls the $L^2$ norm of the context $\xi$ (Kirchmeyer et al., 2022):

$$\Omega_{W,\xi}(W,\xi) = \lambda_W \cdot \sum_{i=1}^{d_\phi} \|W_{i,:}\|_2 + \lambda_\xi \cdot \|\xi\|^2.$$

Analogous to the augmentation method, the source parameters $\phi_0$ are kept fixed, and only $(\theta, W, \xi)$ are optimized.

**Conjugacy with Context Modulation and Adversarial Orthogonality (CM-AO).** Building on context modulation with conjugacy learning, this method further enforces the modulation basis to be orthogonal to the model-induced worst-case orbit-tangent directions by solving the following min-max problem:

$$\min_{\theta,W,\xi} \left[ \mathcal{L}_{\text{conj}}^{\text{flow}}(\theta, W, \xi) + \Omega_\Phi(\theta) + \Omega_{W,\xi}(W, \xi) + \max_{\gamma,\alpha} \left[ \widehat{\mathcal{L}}_{\text{orth}}(W,\gamma,\alpha) - \Omega_v(\gamma,\alpha) \right] \right],$$

Here, $\widehat{\mathcal{L}}_{\text{orth}}(W, \gamma, \alpha)$ is the alignment functional defined in (12), and $\Omega_v(\gamma, \alpha)$ is a $L^2$ complexity term on the model-induced infinitesimal generator, which discourages overly complex generators in the maximization step:

$$\widehat{\mathcal{L}}_{\text{ortho}}(W,\gamma,\alpha) = \lambda_{\text{orth}} \cdot \frac{\left\| \Pi_{\tilde{P}_W} \tilde{r}_{\gamma,\alpha} \right\|^2}{\left\| \tilde{r}_{\gamma,\alpha} \right\|^2}, \qquad \Omega_v(\gamma,\alpha) = \lambda_v \cdot \frac{1}{NT} \sum_{n=1}^{N} \sum_{t=1}^{T} \|v_{\gamma,\alpha}(x_n^t)\|^2.$$

Both $\widehat{\mathcal{L}}_{\text{orth}}$ and $\Omega_v$ constrain the source dynamics and are therefore evaluated in the pullback coordinates. Concretely, the matrices in $\widehat{\mathcal{L}}_{\text{orth}}$ are formed by stacking trajectory-wise evaluations over a batch of samples:

$$\tilde{P}_W = \begin{bmatrix} P_W(x_1^1) \\ \vdots \\ P_W(x_n^t) \\ \vdots \\ P_W(x_N^T) \end{bmatrix} \in \mathbb{R}^{NTd_x \times d_\xi}, \qquad \tilde{r}_{\gamma,\alpha} = \begin{bmatrix} [f_0, v_{\gamma,\alpha}](x_1^1) \\ \vdots \\ [f_0, v_{\gamma,\alpha}](x_n^t) \\ \vdots \\ [f_0, v_{\gamma,\alpha}](x_N^T) \end{bmatrix} \in \mathbb{R}^{NTd_x},$$

where $P_W(x_n^t) = [p_1(x_n^t), \ldots, p_{d_\xi}(x_n^t)] \in \mathbb{R}^{d_x \times d_\xi}$. In addition, we employ a more numerically stable surrogate for $\widehat{\mathcal{L}}_{\text{orth}}$ based on the worst-case trace-ratio alignment; see Appendix D for details.

## D. Worst-case Trace-ratio Alignment Functional

In the main text, we encourage the modulation space $\mathcal{P}_W$ to be orthogonal to the model-induced orbit-tangent set by penalizing the global worst-case alignment

$$\mathcal{L}_{\text{orth}}(\mathcal{P}_W, \mathcal{R}_{\mathcal{A}_\Phi}) = \sup_{r_{\gamma,\alpha} \in \mathcal{R}_{\mathcal{A}_\Phi} \setminus \{0\}} \frac{\|\Pi_{\mathcal{P}_W} r_{\gamma,\alpha}\|^2}{\|r_{\gamma,\alpha}\|^2}, \tag{13}$$

as in (10). To operationalize this, we introduce the min-max objective in (11):

$$\min_{\theta, W, \xi} \max_{\gamma, \alpha} \left[ \mathcal{L}_{\text{conj}}^{\text{flow}}(\theta, W, \xi) + \frac{\|\Pi_{\mathcal{P}_W} r_{\gamma,\alpha}\|^2}{\|r_{\gamma,\alpha}\|^2} \right], \quad v_{\gamma,\alpha} = \sum_{k=1}^{K} \alpha_k v_\gamma^{(k)}, \quad r_{\gamma,\alpha} = [f_0, v_{\gamma,\alpha}] = \sum_{k=1}^{K} \alpha_k [f_0, v_\gamma^{(k)}],$$

which approximates the supremum in (13) by a per-iteration adversarial update that selects a maximizer $(\gamma, \alpha)$. In practice, we found it more stable to replace this objective with a *worst-case trace-ratio alignment* computed over the $K$ layerwise candidates $\{[f_0, v_\gamma^{(k)}]\}_{k=1}^{K}$. This appendix introduces this surrogate as a smooth relaxation of the worst-case alignment within the same realizable orbit-tangent subspace.

**A subspace view of realizable worst-case directions.** For a given $\gamma \in \Gamma$, recall the $K$ realizable orbit-tangent directions:

$$r_\gamma^{(k)} = [f_0, v_\gamma^{(k)}], \qquad k = 1, \ldots, K, \tag{14}$$

and the corresponding at most $K$-dimensional orbit-tangent subspace:

$$T_{f_0}\mathcal{O}_{f_0}(\gamma) = \text{span}\{r_\gamma^{(k)}\}_{k=1}^{K},$$

as introduced in Section 3.4. Clearly, for any $\alpha \in \mathbb{R}^K$, $r_{\gamma,\alpha} = \sum_{k=1}^{K} \alpha_k r_\gamma^{(k)} \in T_{f_0}\mathcal{O}_{f_0}(\gamma)$. Hence, for fixed $\gamma$, the inner supremum over $\alpha$ in (13) is equivalent to the local worst-case alignment over the subspace $T_{f_0}\mathcal{O}_{f_0}(\gamma)$:

$$\mathcal{L}_{\text{orth}}^{(\infty)}(\mathcal{P}_W, \gamma) := \sup_{\alpha \neq 0} \frac{\|\Pi_{\mathcal{P}_W} r_{\gamma,\alpha}\|^2}{\|r_{\gamma,\alpha}\|^2} = \sup_{r \in T_{f_0}\mathcal{O}_{f_0}(\gamma) \setminus \{0\}} \frac{\|\Pi_{\mathcal{P}_W} r\|^2}{\|r\|^2}. \tag{15}$$

The global worst-case alignment is then $\mathcal{L}_{\text{orth}}(\mathcal{P}_W, \mathcal{R}_{\mathcal{A}_\Phi}) = \sup_{\gamma \in \Gamma} \mathcal{L}_{\text{orth}}^{(\infty)}(\mathcal{P}_W, \gamma)$.

**From the local worst-case alignment to a trace-ratio alignment.** The quantity $\mathcal{L}_{\text{orth}}^{(\infty)}(\mathcal{P}_W, \gamma)$ is the local worst-case alignment between $\mathcal{P}_W$ and the realizable orbit-tangent subspace $T_{f_0}\mathcal{O}_{f_0}(\gamma)$ at $\gamma \in \Gamma$. Computing this supremum requires solving an inner maximization over $\alpha$, and its gradient can be unstable when the leading alignment directions are nearly tied. To obtain a stable surrogate while preserving the same geometric intent, we replace this local worst-case alignment by a trace-ratio aggregated over the $K$ realizable directions $\{r_\gamma^{(k)}\}_{k=1}^{K}$ spanning the same subspace. Concretely, we stack the $K$ realizable orbit-tangent directions in (14) as

$$R_\gamma = [r_\gamma^{(1)}, \ldots, r_\gamma^{(K)}].$$

We introduce the following local trace-ratio alignment functional at $\gamma$:

$$\mathcal{L}_{\text{orth}}^{(2)}(\mathcal{P}_W, \gamma) := \frac{\text{Tr}(R_\gamma^\top \Pi_{\mathcal{P}_W} R_\gamma)}{\text{Tr}(R_\gamma^\top R_\gamma)} = \frac{\sum_{k=1}^{K} \|\Pi_{\mathcal{P}_W} r_\gamma^{(k)}\|^2}{\sum_{k=1}^{K} \|r_\gamma^{(k)}\|^2}. \tag{16}$$

It is noteworthy that $\mathcal{L}_{\text{orth}}^{(\infty)}(\mathcal{P}_W, \gamma)$ in (15) is effectively winner-take-all: its gradient is governed by the current maximizer. When two leading directions are nearly tied, the maximizer can flip across minibatches, resulting in high-variance adversarial updates. In contrast, the trace-ratio $\mathcal{L}_{\text{orth}}^{(2)}(\mathcal{P}_W, \gamma)$ in (16) aggregates projection energy across competing directions, so near ties merely redistribute contributions smoothly rather than flipping the active direction. Therefore, this surrogate yields smoother gradients in the min–max dynamics and empirically reduces training oscillations. Moreover, this trace-ratio surrogate also preserves the exact orthogonality condition:

$$\mathcal{L}_{\text{orth}}^{(2)}(\mathcal{P}_W, \gamma) = 0 \iff \Pi_{\mathcal{P}_W} r_\gamma^{(k)} = 0 \; \forall k \iff \Pi_{\mathcal{P}_W} r_{\gamma,\alpha} = 0 \; \forall \alpha \iff \mathcal{L}_{\text{orth}}^{(\infty)}(\mathcal{P}_W, \gamma) = 0 \iff \mathcal{P}_W \perp T_{f_0}\mathcal{O}_{f_0}(\gamma),$$

assuming $\sum_{k=1}^{K} \left\| r_\gamma^{(k)} \right\|^2 > 0$. Thus, the trace-ratio surrogate preserves the same notion of orthogonality as the ideal worst-case objective: at the optimum, both $\mathcal{L}_{\text{orth}}^{(\infty)}(\mathcal{P}_W, \gamma)$ and $\mathcal{L}_{\text{orth}}^{(2)}(\mathcal{P}_W, \gamma)$ enforce $\mathcal{P}_W \perp T_{f_0}\mathcal{O}_{f_0}(\gamma)$ precisely. Finally, replacing the local worst-case alignment (15) with the trace-ratio surrogate (16) and taking the supremum over $\gamma \in \Gamma$ yields the global worst-case trace-ratio alignment:

$$\mathcal{L}_{\text{orth}}^{(2)}(\mathcal{P}_W, \mathcal{R}_{\mathcal{A}_\Phi}) := \sup_{\gamma \in \Gamma} \mathcal{L}_{\text{orth}}^{(2)}(\mathcal{P}_W, \gamma) = \sup_{\gamma \in \Gamma} \frac{\sum_{k=1}^{K} \left\| \Pi_{\mathcal{P}_W} r_\gamma^{(k)} \right\|^2}{\sum_{k=1}^{K} \left\| r_\gamma^{(k)} \right\|^2}. \tag{17}$$

Evaluating the supremum in (17) still relies on adversarial updates over $\gamma \in \Gamma$, whose practical computation is described below.

**Practical computation.** We keep the batchwise stacking notation of the main text. Given a data batch $\{x_b\}_{b=1}^{B}$, stack

$$\tilde{P}_W = \left[ P_W(x_1), \ldots, P_W(x_B) \right]^\top \in \mathbb{R}^{Bd_x \times d_\xi}, \qquad \tilde{r}_\gamma^{(k)} = \left[ r_\gamma^{(k)}(x_1), \ldots, r_\gamma^{(k)}(x_B) \right]^\top \in \mathbb{R}^{Bd_x},$$

where $P_W(x_b) = [p_1(x_b), \ldots, p_{d_\xi}(x_b)] \in \mathbb{R}^{d_x \times d_\xi}$. Stacking the $K$ orbit-tangent directions yields a matrix

$$\tilde{R}_\gamma = \left[ \tilde{r}_\gamma^{(1)}, \ldots, \tilde{r}_\gamma^{(K)} \right] \in \mathbb{R}^{Bd_x \times K}.$$

Using the same ridge-regularized projector as in (12), i.e., $\Pi_{\tilde{P}_W} = \tilde{P}_W (\tilde{P}_W^\top \tilde{P}_W + \lambda I)^{-1} \tilde{P}_W^\top$, we compute the batchwise trace-ratio alignment by

$$\widehat{\mathcal{L}}_{\text{orth}}^{(2)}(W, \gamma) = \frac{\sum_{k=1}^{K} \left\| \Pi_{\mathcal{P}_W} r_\gamma^{(k)} \right\|^2}{\sum_{k=1}^{K} \left\| r_\gamma^{(k)} \right\|^2} \approx \frac{\left\| \Pi_{\tilde{P}_W} \tilde{R}_\gamma \right\|_F^2}{\left\| \tilde{R}_\gamma \right\|_F^2}. \tag{18}$$

This is a drop-in replacement of the single-direction ratio in (12). Unless otherwise specified, we adopt this alignment functional as the experimental baseline for the proposed CM-AO method.

Fig. 10 revisits the Hopf experiment in Fig. 4(a) and tracks the degree of diffeomorphic leakage during training. We quantify diffeomorphic leakage by the squared cosine similarity $\cos^2(\angle(\Delta f_{\text{pred}}, r_{\text{gt}}))$ between the learned correction $\Delta f_{\text{pred}} = f_\xi - f_0$ and the ground-truth orbit-tangent direction $r_{\text{gt}} = [f_0, v_\Phi^{(\text{gt})}]$ in the Hopf setting (Section 4.2). A larger value indicates stronger alignment of the corrective vector field with diffeomorphic variation induced by the observation map. Ideally, adversarial orthogonalization should suppress this leakage.

The green curve corresponds to CM-AO trained with the worst-case trace-ratio alignment in (18), whereas the red curve uses the worst-case single-direction ratio in (12). All other settings and hyperparameters are held fixed. Although

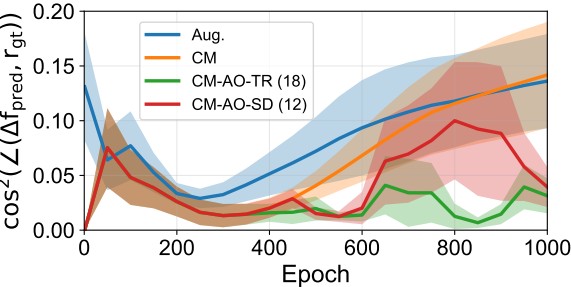

*Figure 10.* **Evolution of alignment metrics during training.** Squared cosine similarity between the learned correction and the ground-truth orbit-tangent direction in the Hopf experiment (Section 4.2). For CM-AO methods (green and red), the orthogonality regularizer is activated after a 300-epoch warm-up period.

both variants reach broadly comparable final performance, the single-direction objective exhibits substantially higher variance throughout training. This empirically supports the trace-ratio alignment as a numerically stable surrogate for the original worst-case direction criterion.

## E. Experimental Details

### E.1. Hopf Bifurcation

**Data generation.** We consider a two-dimensional supercritical Hopf normal form (Marsden & McCracken, 2012) as our benchmark:

$$(\dot{x}_1, \dot{x}_2) = f_\mu^{(\text{gt})}(x_1, x_2) = \left( (\mu - r^2)x_1 - x_2, x_1 + (\mu - r^2)x_2 \right),$$

where $r^2 = x_1^2 + x_2^2$. We set $\mu_s = -0.1$ for the source system and $\mu_t = 0.1$ for the target system. Crossing $\mu = 0$ induces a supercritical Hopf bifurcation: the origin changes from a stable focus to an unstable focus and a stable limit cycle emerges.

We randomly sampled initial conditions uniformly from $[-1, 1]^2$ and integrated trajectories with step size $\Delta t = 0.1$ for $T = 100$ steps. The source dataset consisted of $N_s = 64$ noise-free trajectories integrated from the Hopf normal form $f_{\mu_s}$ with $\mu_s = -0.1$. For the target dataset, we first integrated trajectories $X(t)$ from the Hopf normal form $f_{\mu_t}$ with $\mu_t = 0.1$, then applied a diffeomorphism $\Phi^{(\text{gt})}$ and additive noise:

$$Y(t) = \Phi^{(\text{gt})}(X(t)) + \eta(t), \qquad \eta(t) \sim \mathcal{N}(0, \sigma^2 I),$$

where $\sigma = 0.01$. This dataset is equivalent to integrating trajectories under the pushforward vector field $g_{\text{gt}} = \Phi^{(\text{gt})}_* f^{(\text{gt})}_{\mu_t}$ and then adding noise. We implemented $\Phi^{(\text{gt})}$ as a nonlinear shear-warping map of $(x_1, x_2) \mapsto (x_1, x_2 + s(x_2))$, where the nonlinear shear function $s(\cdot)$ is a sum of Gaussian bumps:

$$s(x_1) = \sum_{m=1}^{M} \beta_m \left[ \exp\left( -\frac{1}{2} \left( \frac{x_1 - a_m}{\sigma_m} \right)^2 \right) - \exp\left( -\frac{1}{2} \left( \frac{-a_m}{\sigma_m} \right)^2 \right) \right],$$

with $M = 2$, $a_m \sim U([-1, 1])$, $\sigma_m \sim U([0.5, 1])$, and $\beta_m$ drawn with random sign and magnitude $|\beta_m| \sim U([1, 2])$. The target dataset contained only a single trajectory ($N_t = 1$) generated by this procedure.

**Common source model training.** The source NODE vector field $f(\cdot; \phi_0)$ was parameterized by a fully connected neural network with three hidden layers, each consisting of 64 hidden units and `swish` activations. We employed 10-roll trajectory training by splitting $N_s = 64$ source trajectories of length $T = 100$ into 640 sub-trajectories of length 10. The source model was then trained using the Adam optimizer (Kingma & Ba, 2015) with a learning rate of $10^{-3}$ and a minibatch size of 64, during 1,000 epochs.

**Target model and training.** Similar to the source training, we employed 10-roll trajectory training by splitting the $N_t = 1$ target trajectory of length $T = 100$ into 10 sub-trajectories of length 10. Each target model was then trained using the Adam optimizer (Kingma & Ba, 2015) with a learning rate of $10^{-3}$ for 1,000 epochs in a full-batch regime. Detailed architectures and hyperparameter settings are provided below:

- **Baseline conjugacy learning.** The conjugacy map $\Phi_\theta$ was parameterized using Real NVP (Dinh et al., 2017) with six alternating affine coupling layers. In each coupling layer, the scale and translation functions were implemented as separate fully connected networks, each with three hidden layers of 64 hidden units and `swish` activations. The scale head outputs a $d_x/2$-dimensional vector with a `tanh` activation, while the translation head outputs a $d_x/2$-dimensional vector with a linear activation. A near-identity regularization term $\Omega_\Phi(\theta)$ was applied with $\lambda_\Phi = 10^{-3}$.

- **Conjugacy with Direct Tuning (DT).** In this setting, the source parameters $\phi_0$ and the conjugacy map parameters $\theta$ were jointly optimized under the conjugacy objective together with the near-identity regularization $\Omega_\Phi(\theta)$, using $\lambda_\Phi = 10^{-3}$. The conjugacy map $\Phi_\theta$ employed the same Real NVP architecture with six affine coupling layers as in the baseline.

- **Conjugacy with $L^2$-SP.** This variant follows the same configuration as conjugacy with direct tuning, with the addition of an $L^2$-SP penalty on the source parameters. The regularization strength was set to $\lambda_{\text{SP}} = 10^{-2}$.

- **Conjugacy with augmentation (Aug.).** For conjugacy with augmentation, the source dynamics were augmented by an additive corrective field, yielding $f_\xi = f_0 + \delta_\xi$, and trained jointly with a conjugacy map $\Phi_\theta$. The corrective field $\delta_\xi$ was parameterized by a fully connected neural network with three hidden layers of 64 hidden units and `swish` activations. The conjugacy map $\Phi_\theta$ followed the same Real NVP architecture as above with $\lambda_\Phi = 10^{-3}$. An $L^2$ complexity penalty $\Omega_\delta(\xi)$ was applied to the corrective field with $\lambda_\delta = 10^{-3}$.

- **Conjugacy with Context Modulation (CM).** In this approach, the source model was adapted via a low-rank context modulation $\phi_0 \mapsto \phi_0 + W\xi$, yielding the modulated vector field $f_\xi(\cdot) = f(\cdot; \phi_0 + W\xi)$, which was trained jointly with a conjugacy map $\Phi_\theta$ using the conjugacy loss. The modulation dimension was set to $d_\xi = 2$, exceeding the ground-truth parameter codimension $d_\mu = 1$ to reflect the absence of prior knowledge. The conjugacy map $\Phi_\theta$ used the same Real NVP architecture with six affine coupling layers and $\lambda_\Phi = 10^{-3}$. A group-Lasso penalty was applied to $W$ with $\lambda_W = 10^{-6}$, and an $L^2$ penalty was applied to $\xi$ with $\lambda_\xi = 10^{-4}$.

- **Conjugacy with Context Modulation and Adversarial Orthogonality (CM-AO).** This method extends conjugacy with context modulation by additionally enforcing adversarial orthogonality against worst-case orbit-tangent directions. As in the context-modulation baseline, the modulation parameters $(W, \xi)$ and the conjugacy map $\Phi_\theta$ were jointly trained under the conjugacy objective, using the same modulation dimension $d_\xi = 2$, the same regularization hyperparameters $\lambda_W = 10^{-6}$, $\lambda_\xi = 10^{-4}$, and $\lambda_\Phi = 10^{-3}$, and the same Real NVP architecture for $\Phi_\theta$ (six affine coupling layers). The adversarial direction field $v_{\gamma,\alpha}$ was parameterized by a separate Real-NVP-based generator with the same coupling architecture (six affine coupling layers). Training alternated one adversary (maximization) update at a learning rate of $2 \times 10^{-4}$ and one main (minimization) update at a learning rate of $10^{-3}$ (Heusel et al., 2017). The orthogonality term was weighted by $\lambda_{\mathrm{orth}} = 10^{-3}$ in the main step and by $\lambda_{\mathrm{orth}}^{\mathrm{adv}} = 1$ in the adversary step. An $L^2$ complexity penalty was applied to the adversarial generator with $\lambda_v = 10^{-3}$. To obtain a reliable initialization of the worst-case orbit-tangent subspace, we employed a 300-epoch initial warm-up phase during which the orthogonality penalty was disabled in the main update, while the adversary continued to be trained. After this warm-up period, $\lambda_{\mathrm{orth}}$ in the main step was gradually increased from zero to its target value of $10^{-3}$ using a cosine ramp schedule.

**Invariant-set metrics.** We approximated the invariant set of each model by uniformly sampling 1,024 initial conditions from $[-1, 1]^2$ and numerically integrating each trajectory for $T = 1000$ steps with a step size $\Delta t = 0.1$. The resulting $(1024, 2)$ point clouds were then used to compute three set-discrepancy metrics between the predicted and ground-truth invariant sets: (i) the Hausdorff distance ($d_H$), defined as the bidirectional maximum nearest-neighbor Euclidean distance between two sets; (ii) the 2-Wasserstein distance ($W_2$), computed by solving an optimal bipartite matching problem with squared Euclidean pairwise costs; and (iii) the Maximum Mean Discrepancy (MMD), evaluated using a Radial Basis Function (RBF) kernel with an $U$-statistic estimator, where the kernel bandwidth is chosen by the median heuristic.

**Target-Only Benchmark.** As a control experiment, we conducted a target-only benchmark for the Hopf experiment. We trained a NODE directly on the target observations, without using any source dynamics and conjugacy-based transfer. We varied the number of target trajectories from $N_t = 1$, corresponding to the original benchmark setting, to $N_t = 5$. Since the target-only baseline does not involve an explicit source dynamics or normal-form identification, we report only invariant-set metrics evaluated in the target observation space.

*Table 4.* Comparison of observation-space invariant-set metrics between CM-AO and target-only NODE baselines on the Hopf benchmark in Section 4.2. Here, $N_t$ denotes the number of target trajectories used for training.

| Method | $d_H \downarrow$ | $W_2 \downarrow$ | MMD $\downarrow$ |
|---|---|---|---|
| Target-only ($N_t = 1$) | $0.305 \pm 0.210$ | $0.275 \pm 0.110$ | $0.523 \pm 0.250$ |
| Target-only ($N_t = 2$) | $0.229 \pm 0.063$ | $0.201 \pm 0.092$ | $0.330 \pm 0.089$ |
| Target-only ($N_t = 3$) | $0.254 \pm 0.236$ | $0.224 \pm 0.101$ | $0.397 \pm 0.255$ |
| Target-only ($N_t = 4$) | $0.071 \pm 0.026$ | $0.169 \pm 0.063$ | $0.246 \pm 0.086$ |
| Target-only ($N_t = 5$) | $\mathbf{0.059 \pm 0.028}$ | $\underline{0.122 \pm 0.027}$ | $\underline{0.188 \pm 0.061}$ |
| CM-AO ($N_t = 1$) | $\underline{0.064 \pm 0.007}$ | $\mathbf{0.121 \pm 0.020}$ | $\mathbf{0.182 \pm 0.037}$ |

As shown in Table 4, the target-only baseline improves as more target trajectories are provided, but even with $N_t = 5$, its performance is only comparable to CM-AO trained with a single target trajectory. This highlights the sample-efficiency advantage of the proposed conjugacy-based transfer framework. By leveraging structural information inherited from the source normal form, together with a small topological correction, CM-AO can recover the underlying canonical dynamics and reproduce the target invariant-set structure in observation space using substantially fewer target trajectories.

### E.2. Duffing Oscillators

**Data generation.** We consider a two-dimensional Duffing oscillator (Strogatz, 2024) as our benchmark:

$$(\dot{x}_1, \dot{x}_2) = f_\mu^{(\mathrm{gt})}(x_1, x_2) = (x_2, \mu x_1 - x_1^3),$$

where $\mu_s = -0.1$ for the source system and $\mu_t = 0.1$ for the target system. Crossing $\mu = 0$ induces a pitchfork-type bifurcation in the associated potential landscape: the system transitions from a single-well (mono-stable) regime to a double-well (bi-stable) regime, where two center equilibria emerge and the origin becomes an unstable saddle point. We randomly sampled initial conditions uniformly from $[-0.5, 0.5]^2$ and integrated trajectories with step size $\Delta t = 0.1$ for $T = 100$ steps. The source dataset consisted of $N_s = 64$ noise-free trajectories integrated from the source normal form $f_{\mu_s}$. For the target dataset, we first integrated trajectories $X(t)$ from the target normal form $f_{\mu_t}$, then applied a diffeomorphism $\Phi^{(\mathrm{gt})}$ and additive noise:

$$Y(t) = \Phi^{(\mathrm{gt})}(X(t)) + \eta(t), \qquad \eta(t) \sim \mathcal{N}(0, \sigma^2 I),$$

with $\sigma = 0.01$. This dataset is equivalent to integrating trajectories under the pushforward vector field $g_{\text{gt}} = \Phi_*^{(\text{gt})} f_{\mu_t}^{(\text{gt})}$ and then adding noise. We implemented $\Phi^{(\text{gt})}$ as a composition of a smooth parabolic shear and a global affine map:

$$\Phi^{(\text{gt})}(x) = A w(x) + t, \qquad w(x_1, x_2) = \big( x_1, x_2 + \kappa x_1^2 \big),$$

where $\kappa$ controls the strength of the symmetry-breaking deformation. The target dataset consisted of $N_t = 4$ noisy trajectories produced by this procedure. Within each trial, the parameters of $\Phi^{(\text{gt})}$ were fixed and shared across the four generated target trajectories; across trials, they were resampled by drawing a rotation angle $\theta \sim U([-0.5, 0.5])$ (in radians), a global scale $c \sim U([0.8, 1.2])$, and a linear shear $\rho \sim U([-0.2, 0.2])$, and setting $A = R(\theta)\, S(c)\, \text{Sh}(\rho)$. For the nonlinear warp, we sampled the curvature coefficient as $\kappa = sm$, where $s = \text{sign}(u)$ with $u \sim U([-1, 1])$ and $m \sim U([0.5, 1.5])$.

**Common source model training.** The source NODE vector field $f(\cdot; \phi_0)$ was parameterized by a fully connected neural network with three hidden layers, each consisting of 64 hidden units and `swish` activations. We employed 10-roll trajectory training by splitting $N_s = 64$ source trajectories of length $T = 100$ into 640 sub-trajectories of length 10. The source model was then trained using the Adam optimizer (Kingma & Ba, 2015) with a learning rate of $10^{-3}$ and a minibatch size of 64, during 1,000 epochs.

**Target model and training.** Similar to the source training, we employed 10-roll trajectory training by splitting the $N_t = 4$ target trajectory of length $T = 100$ into 40 sub-trajectories of length 10. Each target model was then trained using the Adam optimizer (Kingma & Ba, 2015) with a learning rate of $10^{-3}$ for 1,000 epochs in a full-batch regime. Detailed architectures and hyperparameter settings are provided below:

- **Baseline conjugacy learning.** The conjugacy map $\Phi_\theta$ was parameterized using Real NVP (Dinh et al., 2017) with six alternating affine coupling layers. In each coupling layer, the scale and translation functions were implemented as separate fully connected networks, each with three hidden layers of 64 hidden units and `swish` activations. The scale head outputs a $d_x/2$-dimensional vector with a `tanh` activation, while the translation head outputs a $d_x/2$-dimensional vector with a linear activation. A near-identity regularization term $\Omega_\Phi(\theta)$ was applied with $\lambda_\Phi = 10^{-3}$.

- **Conjugacy with Direct Tuning (DT).** In this setting, the source parameters $\phi_0$ and the conjugacy map parameters $\theta$ were jointly optimized under the conjugacy objective together with the near-identity regularization $\Omega_\Phi(\theta)$, using $\lambda_\Phi = 10^{-3}$. The conjugacy map $\Phi_\theta$ employed the same Real NVP architecture with six affine coupling layers as in the baseline.

- **Conjugacy with $L^2$-SP.** This variant follows the same configuration as conjugacy with direct tuning, with the addition of an $L^2$-SP penalty on the source parameters. The regularization strength was set to $\lambda_{\text{SP}} = 10^{-2}$.

- **Conjugacy with augmentation (Aug.).** For conjugacy with augmentation, the source dynamics were augmented by an additive corrective field, yielding $f_\xi = f_0 + \delta_\xi$, and trained jointly with a conjugacy map $\Phi_\theta$. The corrective field $\delta_\xi$ was parameterized by a fully connected network with three hidden layers of 64 hidden units and `swish` activations. The conjugacy map $\Phi_\theta$ followed the same Real NVP architecture as above with $\lambda_\Phi = 10^{-3}$. An $L^2$ complexity penalty $\Omega_\delta(\xi)$ was applied to the corrective field with $\lambda_\delta = 10^{-3}$.

- **Conjugacy with Context Modulation (CM).** In this approach, the source model was adapted via a low-rank context modulation $\phi_0 \mapsto \phi_0 + W\xi$, yielding the modulated vector field $f_\xi(\cdot) = f(\cdot; \phi_0 + W\xi)$, which was trained jointly with a conjugacy map $\Phi_\theta$ using the conjugacy loss. The modulation dimension was set to $d_\xi = 6$, exceeding the ground-truth parameter codimension $d_\mu = 1$ to reflect the absence of prior knowledge. The conjugacy map $\Phi_\theta$ used the same Real NVP architecture with six affine coupling layers and $\lambda_\Phi = 10^{-3}$. A group-Lasso penalty was applied to $W$ with $\lambda_W = 10^{-6}$, and an $L^2$ penalty was applied to $\xi$ with $\lambda_\xi = 10^{-4}$.

- **Conjugacy with Context Modulation and Adversarial Orthogonality (CM-AO).** This method extends conjugacy with context modulation by additionally enforcing adversarial orthogonality against worst-case orbit-tangent directions. As in the context-modulation baseline, the modulation parameters $(W, \xi)$ and the conjugacy map $\Phi_\theta$ were jointly trained under the conjugacy objective, using the same modulation dimension $d_\xi = 6$, the same regularization hyperparameters $\lambda_W = 10^{-6}$, $\lambda_\xi = 10^{-4}$, and $\lambda_\Phi = 10^{-3}$, and the same Real NVP architecture for $\Phi_\theta$ (six affine coupling layers). The adversarial direction field $v_{\gamma,\alpha}$ was parameterized by a separate Real-NVP-based generator with the same coupling architecture (six affine coupling layers). Training alternated one adversary (maximization) update at a learning rate of

$10^{-3}$ and one main (minimization) update at a learning rate of $10^{-3}$ (Heusel et al., 2017). The orthogonality term was weighted by $\lambda_{\text{orth}} = 10^{-3}$ in the main step and by $\lambda_{\text{orth}}^{\text{adv}} = 1$ in the adversary step. An $L^2$ complexity penalty was applied to the adversarial generator with $\lambda_v = 10^{-3}$. To obtain a reliable initialization of the worst-case orbit-tangent subspace, we employed a 300-epoch initial warm-up phase during which the orthogonality penalty was disabled in the main update, while the adversary continued to be trained. After this warm-up period, $\lambda_{\text{orth}}$ in the main step was gradually increased from zero to its target value of $10^{-3}$ using a cosine ramp schedule.

We applied the orthogonality regularization both to the pullbacked observations, $x = \Phi_\theta^{-1}(y)$, and to samples drawn from a predefined distribution $x \sim \mathcal{N}(x_0, \sigma^2)$, where $x_0$ is an equilibrium of the source model $f_0$ (i.e., $f_0(x_0) = 0$) and $\sigma = 0.1$. This additional sampling regularizes the modulation in a neighborhood of the equilibrium, which is critical for preserving the system's local topology.

**Symmetry and MSE metrics.** We quantified $\mathbb{Z}_2$-symmetry violation using the relative error defined as

$$\text{Score}_{\text{sym}} := \frac{\sqrt{\mathbb{E}\|f_{\text{pred}}(x) + f_{\text{pred}}(-x)\|^2}}{\sqrt{\mathbb{E}\|f_{\text{pred}}(x)\|^2}},$$

computed for each corrected source model $f_{\text{pred}}$. The Mean Squared Error (MSE) was defined in the normal-form coordinates as $\mathbb{E}\big[\|f_{\text{pred}}(x) - f_{\mu_t}^{(\text{gt})}(x)\|^2\big]$ and in the observational coordinates as $\mathbb{E}\big[\|(\Phi_\theta)_* f_{\text{pred}}(y) - g_{\text{gt}}(y)\|^2\big]$, where $g_{\text{gt}} = \Phi_*^{(\text{gt})} f_{\mu_t}^{(\text{gt})}$. For both metrics, expectations were estimated over a meshgrid on $[-0.5, 0.5]^2$ with $\Delta x = 0.025$.

### E.3. Hopf Bifurcation with Phase-Space Extrapolation

**Data generation.** We used the same supercritical Hopf bifurcation system, observation diffeomorphism (the nonlinear shear-warping map), and Gaussian noise setting as in Appendix E.1, changing only the target initial-condition distribution and the roll-out horizon. Specifically, target trajectories were sampled only from initial conditions far from the limit cycle, with $r \sim [1, 1.2]$ compared with the limit-cycle radius $\sqrt{\mu} \approx 0.32$, and were observed over a short horizon of $T = 20$ with $\Delta t = 0.1$. The resulting trajectories were then transformed by the diffeomorphism and corrupted with Gaussian noise, as shown in Fig. 11. Consequently, the training data did not directly reveal the limit-cycle geometry. We used eight target trajectories $N_t = 8$ in this experiment.

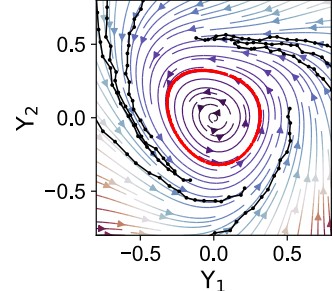

*Figure 11.* **Hopf dynamics in the extrapolative setting.** Black curves show the observed trajectories, while the red curve indicates the limit cycle.

**Self-regularization.** In this benchmark, the observed trajectories do not reach the relevant topological structure. Therefore, applying the orthogonality regularization only on the observed data primarily constrains the outer phase-space region, rather than the topologically meaningful equilibrium structure. To address this limitation, we introduce a simple self-regularization strategy: during training, we randomly sample initial points from $[x_{\min}, x_{\max}]^2$, roll out the learned CM-AO dynamics $f_\xi$ over a longer horizon $T_{\text{aug}}$, and apply the orthogonality regularization also on the resulting $N_{\text{aug}}$ trajectories. We used $x_{\min} = -1$, $x_{\max} = 1$, $T_{\text{aug}} = 100$, and $N_{\text{aug}} = 8$.

### E.4. FitzHugh–Nagumo Reaction–Diffusion

**Data generation.** We consider the FitzHugh–Nagumo Partial Differential Equation (PDE) (Rocsoreanu et al., 2012):

$$\partial_t u = D_u \partial_x^2 u + u - \frac{u^3}{3} - v + \mu, \qquad \partial_t v = \zeta(u + a - bv),$$

with periodic boundary conditions. After spatial discretization on a uniform grid, we obtain $2n_x$-dimensional ODE:

$$\dot{u}_j = D_u \Delta_h u_j + u_j - \frac{u_j^3}{3} - v_j + \mu, \qquad \dot{v}_j = \zeta(u_j + a - bv_j),$$

where $j = 1, ..., n_x$. Here, $\Delta_h$ denotes the periodic finite-difference Laplacian (with grid spacing $h = 1$), and $x = (u_1, v_1, \ldots, u_{n_x}, v_{n_x}) \in \mathbb{R}^{2n_x}$. We set $n_x = 32$, $\zeta = 0.08$, $a = 0.7$, $b = 0.9$, and $D_u = 0.15$. The source parameter $\mu_s = 0.0$ yields a quiescent resting regime in which trajectories converge to a homogeneous fixed point, whereas the target parameter $\mu_t = 0.6$ induces traveling-wave spatiotemporal oscillations.

Source trajectories were generated from randomly sampled initial conditions in $[-1, 1]^{2n_x}$. Target trajectories were generated from phase-controlled sinusoidal initial conditions. Specifically, for each target trajectory, we sampled a random phase $\varphi \sim U([0, 2\pi])$ and set

$$\theta_j = \frac{2\pi j}{n_x}, \qquad u_j(0) = A\cos(\omega\theta_j + \varphi) + \eta_j^u, \qquad v_j(0) = 0.8A\sin(\omega\theta_j + \varphi) + \eta_j^v,$$

where $A = 1$ and $\eta_j^u, \eta_j^v \sim \mathcal{N}(0, 0.05^2)$. Here, $\omega$ denotes the spatial phase mode of the initial condition: over one period of the spatial grid, the phase of the sinusoidal profile winds $\omega$ times. We fixed $\omega = 1$ for target training trajectories and used additional values of $\omega$ for evaluation, as discussed later. The resulting target trajectories were then transformed by a fixed observation diffeomorphism. The observation map acts nodewise as

$$\tilde{u}_j = a_j^u u_j + c_j^u, \qquad \tilde{v}_j = a_j^v\big(v_j + \beta_j\tanh(\gamma_j u_j) + \kappa_j u_j^2\big) + c_j^v,$$

where the coefficients are fixed smooth functions of the spatial index:

$$\begin{aligned}
a_j^u &= 1 + 0.08\sin(\theta_j + 0.3), & a_j^v &= 1 + 0.18\cos(\theta_j - 0.2), \\
c_j^u &= 0.08\sin(\theta_j + 1.1), & c_j^v &= 0.18\cos(\theta_j + 0.5), \\
\beta_j &= 0.35\sin(\theta_j + 0.7), & \kappa_j &= 0.1 + 0.06\cos(2\theta_j - 0.4), \\
\gamma_j &= 1.1 + 0.25\sin(\theta_j - 0.8). &&
\end{aligned}$$

The transformed trajectories were further corrupted with additive Gaussian noise, $\eta(t) \sim \mathcal{N}(0, \sigma^2 I)$, with $\sigma = 0.01$. Unless otherwise specified, we used $N_s = 64$ source trajectories and a single target trajectory ($N_t = 1$), rolled out for $T = 1000$ and $T = 500$ steps, respectively, with time step $\Delta t = 0.05$.

**Common source model training.** The source NODE vector field $f(\cdot; \phi_0)$ was parameterized by a convolutional neural network defined on the $(n_x, 2)$ spatial-channel layout. The network first projects the two input channels into a hidden representation, followed by two periodic one-dimensional convolutional layers with kernel size 3, hidden dimension 96, and `swish` activations. A final pointwise linear projection maps the hidden representation back to the two-channel vector field. Periodic padding was used in all convolutional layers to match the periodic boundary condition of the considered system.

We pretrained the source NODE on the source trajectories using standard trajectory matching. The source dataset consisted of $N_s = 64$ trajectories of length $T = 1000$ with $\Delta t = 0.05$, which were split into sub-trajectories of length 20. The source model was trained using the Adam optimizer (Kingma & Ba, 2015) with a learning rate of $10^{-3}$ and a minibatch size of 64, during 1,000 epochs. The optimized parameters define the common source model $f_0(\cdot) = f(\cdot; \phi_0)$, which was used to initialize all target-side transfer learning methods.

**Target model and training.** The target dataset consisted of a single observed target trajectory ($N_t = 1$) of length $T = 500$ with time step $\Delta t = 0.05$, which was split into sub-trajectories of length 10. Each target model was trained in a full-batch regime for 2,000 epochs using the Adam optimizer (Kingma & Ba, 2015) with learning rate $10^{-3}$. Detailed architectures and hyperparameter settings are provided below.

- **Baseline conjugacy learning.** The conjugacy map $\Phi_\theta$ was parameterized by a structured Real NVP (Dinh et al., 2017) architecture adapted to the spatial PDE layout. Internally, the flattened state was rearranged into separated $u$- and $v$-blocks. The $u$-block was transformed by a nodewise affine map, while the $v$-block was transformed by an affine coupling layer conditioned on the transformed $u$-block. The conditioner was implemented using periodic one-dimensional convolutions with hidden dimension 64, kernel size 3, two convolutional layers, `swish` activations, and sinusoidal positional encoding. The scale outputs were clipped using a tanh parameterization with scale factor 0.25. A near-identity regularization term $\Omega_\Phi(\theta)$ was applied with $\lambda_\Phi = 10^{-5}$.

- **Conjugacy with Direct Tuning (DT).** In this setting, the source parameters $\phi_0$ and the conjugacy map parameters $\theta$ were jointly optimized under the conjugacy objective together with the near-identity regularization $\Omega_\Phi(\theta)$, using $\lambda_\Phi = 10^{-5}$. The conjugacy map $\Phi_\theta$ employed the same structured Real NVP architecture as in the baseline

- **Conjugacy with $L^2$-SP.** This variant follows the same configuration as conjugacy with direct tuning, with the addition of an $L^2$-SP penalty on the source parameters. The regularization strength was set to $\lambda_{\text{SP}} = 10^{-2}$.

- **Conjugacy with augmentation (Aug.).** For conjugacy with augmentation, the source dynamics were augmented by an additive corrective field, yielding $f_\xi = f_0 + \delta_\xi$, and trained jointly with a conjugacy map $\Phi_\theta$. The corrective field $\delta_\xi$ was parameterized by a convolutional neural network with the same spatial layout as the source NODE, using periodic one-dimensional convolutions. The conjugacy map $\Phi_\theta$ followed the same structured Real NVP architecture as above with $\lambda_\Phi = 10^{-5}$. An $L^2$ complexity penalty $\Omega_\delta(\xi)$ was applied to the corrective field with $\lambda_\delta = 10^{-3}$.

- **Conjugacy with Context Modulation (CM).** In this approach, the source model was adapted via a low-rank context modulation $\phi_0 \mapsto \phi_0 + W\xi$, yielding the modulated vector field $f_\xi(\cdot) = f(\cdot; \phi_0 + W\xi)$, which was trained jointly with a conjugacy map $\Phi_\theta$ using the conjugacy loss. The modulation dimension was set to $d_\xi = 6$, exceeding the ground-truth parameter codimension $d_\mu = 1$ to reflect the absence of prior knowledge. The conjugacy map $\Phi_\theta$ used the same structured Real NVP architecture and $\lambda_\Phi = 10^{-5}$. A group-Lasso penalty was applied to $W$ with $\lambda_W = 10^{-6}$, and an $L^2$ penalty was applied to $\xi$ with $\lambda_\xi = 10^{-4}$.

- **Conjugacy with Context Modulation and Adversarial Orthogonality (CM-AO).** This method extends conjugacy with context modulation by additionally enforcing adversarial orthogonality against worst-case orbit-tangent directions. As in the context-modulation baseline, the modulation parameters $(W, \xi)$ and the conjugacy map $\Phi_\theta$ were jointly trained under the conjugacy objective, using the same modulation dimension $d_\xi = 6$, the same regularization hyperparameters $\lambda_W = 10^{-6}$, $\lambda_\xi = 10^{-4}$, and $\lambda_\Phi = 10^{-5}$, and the same structured Real NVP architecture for $\Phi_\theta$. The adversarial direction field $v_{\gamma,\alpha}$ was parameterized by a separate Real-NVP-based generator with the same coupling architecture. Training alternated one adversary (maximization) update at a learning rate of $2 \times 10^{-4}$ and one main (minimization) update at a learning rate of $10^{-3}$ (Heusel et al., 2017). The orthogonality term was weighted by $\lambda_{\mathrm{orth}} = 10^{-3}$ in the main step and by $\lambda_{\mathrm{orth}}^{\mathrm{adv}} = 1$ in the adversary step. An $L^2$ complexity penalty was applied to the adversarial generator with $\lambda_v = 10^{-3}$. To obtain a reliable initialization of the worst-case orbit-tangent subspace, we employed a 500-epoch initial warm-up phase during which the orthogonality penalty was disabled in the main update, while the adversary continued to be trained. After this warm-up period, $\lambda_{\mathrm{orth}}$ in the main step was gradually increased from zero to its target value of $10^{-3}$ using a cosine ramp schedule.

**Invariant-set and trajectory MSE metrics.** Because the invariant set of the discretized reaction–diffusion system lies in a high-dimensional phase space ($2n_x = 64$), we evaluated invariant-set metrics using a Monte Carlo approximation. For each evaluation, we sampled $N = 2000$ initial conditions from the same phase-controlled sinusoidal family used for target data generation, rolled out both the ground-truth target system and each learned model for $T = 2000$ steps, and collected the final $T_{\mathrm{used}} = 50$ states as point-cloud approximations of the long-time invariant set. We then compared the predicted and ground-truth point clouds in the full $2n_x$-dimensional state space using the Hausdorff distance ($d_H$), 2-Wasserstein distance ($W_2$), and MMD as in Appendix E.1. For computational efficiency, each metric was computed on randomly subsampled point clouds of size $(2000, 64)$. This procedure was repeated over five random subsamplings and averaged.

In addition to these invariant-set metrics, we report the standard trajectory MSE between ground-truth and predicted rollouts. This metric directly measures spatiotemporal rollout fidelity from matched initial conditions, making it particularly useful for comparing traveling-wave patterns in high-dimensional state spaces.

We evaluated two spatial phase modes $\omega$ in the initial conditions. The $\omega = 1$ setting matches the phase mode used for the target training trajectories, serving as an interpolation test. By contrast, the $\omega = 2$ setting initializes the system with a different spatial winding pattern, defining a phase-space extrapolation test. This setting probes whether the learned model recovers the long-time dynamical structure beyond the observed initial-condition regime.

## F. Additional Experimental Results and Visualizations

### F.1. Hopf Bifurcation

Fig. 12 shows one representative example of the learned target dynamics in observation space, $(\Phi_\theta)_*(f_{\mathrm{pred}})$, for the Hopf benchmark. We compare (a) baseline conjugacy learning ($f_{\mathrm{pred}} = f_0$), (b) DT ($f_{\mathrm{pred}} = f_\phi$, $\phi \leftarrow \phi_0$), (c) $L^2$-SP ($f_{\mathrm{pred}} = f_\phi$, $\phi \approx \phi_0$), (d) augmentation ($f_{\mathrm{pred}} = f_0 + \delta_\xi$), (e) CM ($f_{\mathrm{pred}} = f(\cdot; \phi_0 + W\xi)$), and (f) CM-AO ($f_{\mathrm{pred}} = f(\cdot; \phi_0 + W\xi)$, $\mathcal{P}_W \to \mathcal{P}_\perp$). Ideally, these models should reproduce the target observational system $g_{\mathrm{gt}} = \Phi_*^{(\mathrm{gt})} f_{\mu_t}^{(\mathrm{gt})}$ and its invariant set shown in Fig. 2(c). The baseline conjugacy approach fails to recover the correct limit-cycle invariant set because of the topological mismatch between the source and target systems. By contrast, all approaches that explicitly modify the source dynamics successfully reproduce the correct limit-cycle topology in the observed coordinates.

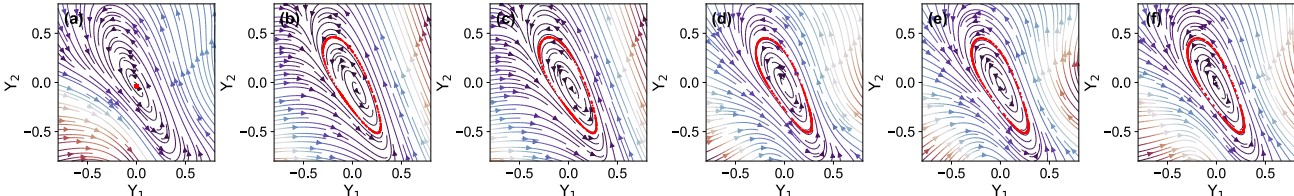

*Figure 12.* **Phase portraits of the learned observational dynamics, $(\Phi_\theta)_*(f_{\mathrm{pred}})$, for a Hopf bifurcating system.** (a) Baseline conjugacy learning. (b) Direct Tuning (DT). (c) $L^2$-SP. (d) Augmentation. (e) Context Modulation (CM). (f) Context Modulation with Adversarial Orthogonality (CM-AO).

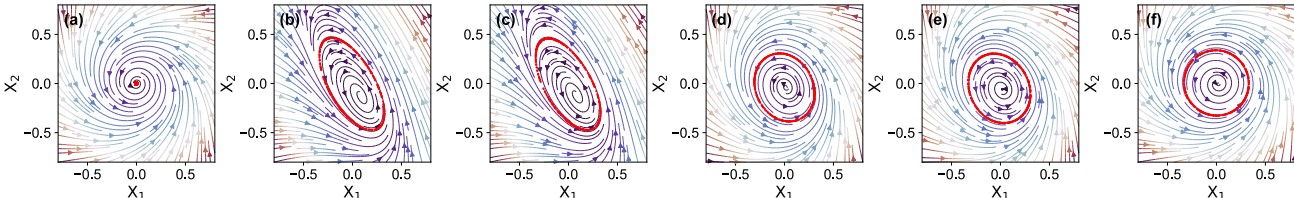

*Figure 13.* **Phase portraits of the learned corrected dynamics, $f_{\mathrm{pred}}$, for a Hopf bifurcating system.** (a) Baseline conjugacy learning (which coincides with the uncorrected source model). (b) Direct Tuning (DT). (c) $L^2$-SP. (d) Augmentation. (e) Context Modulation (CM). (f) Context Modulation with Adversarial Orthogonality (CM-AO).

Within this observational space, the performance differences among these correction-based approaches remain marginal, consistent with the quantitative results reported in Table 1.

However, these seemingly minimal differences among correction-based models become clearly distinguishable when examining the underlying source-correction dynamics $f_{\mathrm{pred}}$ itself. Fig. 13 compares the phase portraits of the corrected source dynamics learned by (a) baseline conjugacy learning (which coincides with the uncorrected source model), (b) DT, (c) $L^2$-SP, (d) augmentation, (e) CM, and (f) CM-AO. Ideally, the corrected dynamics should reproduce the target normal form $f_{\mu_t}^{(\mathrm{gt})}$ shown in Fig. 2(b). For the competing methods in (b–e), the learned vector fields and the resulting limit cycles remain visibly deformed, indicating that the corrections retain residual diffeomorphic components. In contrast, the phase portrait produced by the proposed CM-AO method in (f) produces a nearly $\mathrm{SO}(2)$-symmetric vector field and a circular limit cycle that closely matches the ground truth. This suggests that the orthogonality constraint effectively suppresses diffeomorphic variations, yielding the minimal correction required to compensate the topological discrepancy.

### F.2. Duffing Oscillators

Fig. 14 shows one representative example of the learned target dynamics in observation space, $(\Phi_\theta)_*(f_{\mathrm{pred}})$, for the Duffing benchmark. We compare (a) baseline conjugacy learning ($f_{\mathrm{pred}} = f_0$), (b) DT ($f_{\mathrm{pred}} = f_\phi$, $\phi \leftarrow \phi_0$), (c) $L^2$-SP ($f_{\mathrm{pred}} = f_\phi$, $\phi \approx \phi_0$), (d) augmentation ($f_{\mathrm{pred}} = f_0 + \delta_\xi$), (e) CM ($f_{\mathrm{pred}} = f(\cdot; \phi_0 + W\xi)$), and (f) CM-AO ($f_{\mathrm{pred}} = f(\cdot; \phi_0 + W\xi)$, $\mathcal{P}_W \to \mathcal{P}_\perp$). Ideally, these models should reproduce the target observational double-well system $g_{\mathrm{gt}} = \Phi_*^{(\mathrm{gt})} f_{\mu_t}^{(\mathrm{gt})}$ shown in Fig. 6(c). The baseline conjugacy approach fails to recover the correct double-well structure because of the topological mismatch between the source and target systems. By contrast, all approaches that explicitly modify the source dynamics successfully reproduce the correct double-well topology in the observed coordinates. Within this observational space, the performance differences among these correction-based approaches remain marginal, consistent with the quantitative results reported in Fig. 7(c).

However, these seemingly minimal differences among correction-based models become clearly distinguishable when examining the underlying source-correction dynamics $f_{\mathrm{pred}}$ itself. Fig. 15 compares the phase portraits of the corrected source dynamics learned by (a) baseline conjugacy learning (which coincides with the uncorrected source model), (b) DT, (c) $L^2$-SP, (d) augmentation, (e) CM, and (f) CM-AO. Ideally, the corrected dynamics should reproduce the $\mathbb{Z}_2$-equivariant target normal form $f_{\mu_t}^{(\mathrm{gt})}$ shown in Fig. 6(b). For the competing methods in (b–e), the learned vector fields remain visibly deformed, shifted, or asymmetric, indicating that the corrections retain residual diffeomorphic components. In contrast, the phase portrait produced by the proposed method in (f) produces a nearly $\mathbb{Z}_2$-equivariant vector field with a symmetric double-well potential that closely matches the ground truth. This suggests that the orthogonality constraint effectively suppresses diffeomorphic variations, yielding the minimal correction required to compensate the topological discrepancy.

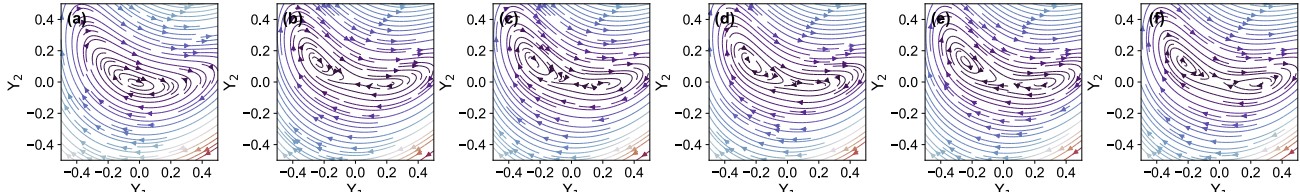

*Figure 14.* **Phase portraits of the learned observational dynamics, $(\Phi_\theta)_*(f_{\mathrm{pred}})$, for a Duffing oscillator.** (a) Baseline conjugacy learning. (b) Direct Tuning (DT). (c) $L^2$-SP. (d) Augmentation. (e) Context Modulation (CM). (f) Context Modulation with Adversarial Orthogonality (CM-AO).

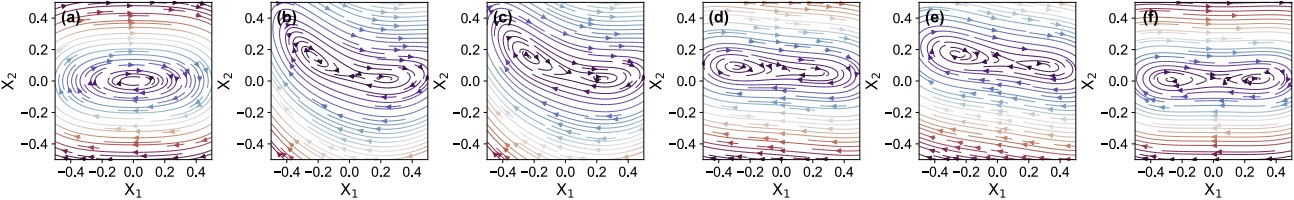

*Figure 15.* **Phase portraits of the learned corrected dynamics, $f_{\mathrm{pred}}$, for a Duffing oscillator.** (a) Baseline conjugacy learning (which coincides with the uncorrected source model). (b) Direct Tuning (DT). (c) $L^2$-SP. (d) Augmentation. (e) Context Modulation (CM). (f) Context Modulation with Adversarial Orthogonality (CM-AO).

## F.3. Hopf Bifurcation with Phase-Space Extrapolation

Fig. 16 shows one representative example of the learned target dynamics in observation space, $(\Phi_\theta)_*(f_{\mathrm{pred}})$, for the Hopf extrapolation benchmark. We compare (a) baseline conjugacy learning ($f_{\mathrm{pred}} = f_0$), (b) DT ($f_{\mathrm{pred}} = f_\phi$, $\phi \leftarrow \phi_0$), (c) $L^2$-SP ($f_{\mathrm{pred}} = f_\phi$, $\phi \approx \phi_0$), (d) augmentation ($f_{\mathrm{pred}} = f_0 + \delta_\xi$), (e) CM ($f_{\mathrm{pred}} = f(\cdot; \phi_0 + W\xi)$), and (f) CM-AO ($f_{\mathrm{pred}} = f(\cdot; \phi_0 + W\xi)$, $\mathcal{P}_W \to \mathcal{P}_\perp$). Ideally, these models should reproduce the target observational system $g_{\mathrm{gt}} = \Phi_*^{(\mathrm{gt})} f_{\mu_t}^{(\mathrm{gt})}$ and its invariant set shown in Fig. 11. The results show that the proposed CM-AO method recovers a limit-cycle geometry close to the ground truth, whereas the other methods either fail to generate a stable limit cycle or overestimate/underestimate its radius. Fig. 17 further compares the phase portraits of the underlying corrected source dynamics $f_{\mathrm{pred}}$ itself, learned by the same six methods. The proposed method in (f) yields a nearly $\mathrm{SO}(2)$-symmetric vector field with a circular limit cycle that closely matches the ground-truth target normal-form dynamics shown in Fig. 2(b).

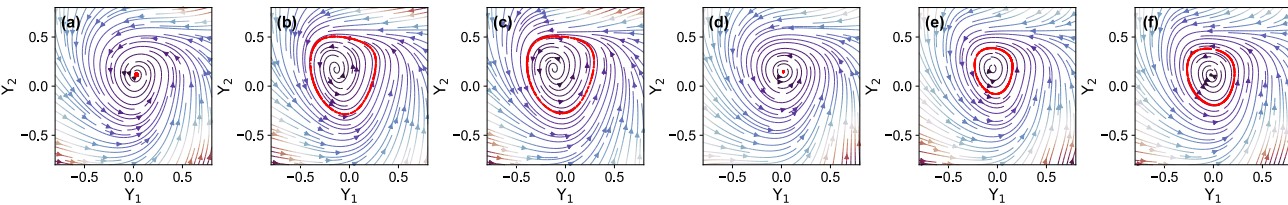

*Figure 16.* **Phase portraits of the learned observational dynamics, $(\Phi_\theta)_*(f_{\mathrm{pred}})$, for the Hopf extrapolation benchmark.** (a) Baseline conjugacy learning. (b) Direct Tuning (DT). (c) $L^2$-SP. (d) Augmentation. (e) Context Modulation (CM). (f) Context Modulation with Adversarial Orthogonality (CM-AO)

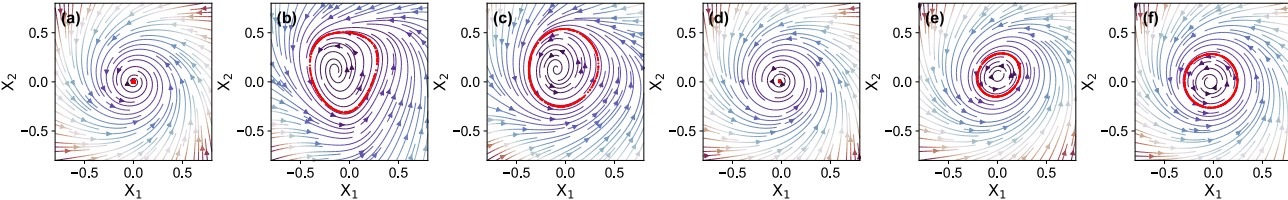

*Figure 17.* **Phase portraits of the learned corrected dynamics, $f_{\mathrm{pred}}$, for the Hopf extrapolation benchmark.** (a) Baseline conjugacy learning (which coincides with the uncorrected source model). (b) Direct Tuning (DT). (c) $L^2$-SP. (d) Augmentation. (e) Context Modulation (CM). (f) Context Modulation with Adversarial Orthogonality (CM-AO).

*Table 5.* FitzHugh–Nagumo benchmark results in observation space. Lower is better. Repeated five times.

| Method | $\omega = 1$ (Interpolation) | | | | $\omega = 2$ (Extrapolation) | | | |
|---|---|---|---|---|---|---|---|---|
| | $d_H$ | $W_2$ | MMD | MSE | $d_H$ | $W_2$ | MMD | MSE |
| Baseline | 11.81 | 10.51 | 0.812 | 1.827 | 11.76 | 11.25 | 0.827 | 1.821 |
| DT | 144.1 | 111.6 | 0.361 | 159.9 | 26.35 | 11.33 | 0.360 | 4.602 |
| $L^2$-SP | 180.7 | 158.0 | 0.356 | 308.6 | 86.20 | 23.04 | 0.350 | 6.183 |
| Aug. | 8.773 | 7.967 | 0.463 | 1.986 | **9.129** | 4.634 | 0.190 | 1.362 |
| CM | 6.996 | 4.810 | 0.214 | 0.194 | 10.21 | 7.735 | 0.440 | 0.721 |
| CM-AO | **4.637** | **3.910** | **0.173** | **0.170** | 9.349 | **2.908** | **0.130** | **0.212** |

*Table 6.* FitzHugh–Nagumo benchmark results in the canonical $(u, v)$ coordinates. Lower is better. Repeated five times.

| Method | $\omega = 1$ (Interpolation) | | | | $\omega = 2$ (Extrapolation) | | | |
|---|---|---|---|---|---|---|---|---|
| | $d_H$ | $W_2$ | MMD | MSE | $d_H$ | $W_2$ | MMD | MSE |
| Baseline | 12.29 | 11.87 | 0.848 | 2.199 | 12.92 | 12.59 | 0.869 | 2.198 |
| DT | 160.0 | 123.3 | 0.304 | 199.0 | 29.70 | 12.46 | 0.312 | 6.080 |
| $L^2$-SP | 204.1 | 181.1 | 0.297 | 408.9 | 97.24 | 26.10 | 0.306 | 8.119 |
| Aug. | 9.717 | 9.202 | 0.590 | 2.218 | 9.725 | 6.114 | 0.369 | 1.513 |
| CM | 6.993 | 4.949 | 0.242 | 0.199 | 10.38 | 7.849 | 0.469 | 0.703 |
| CM-AO | **4.644** | **4.034** | **0.198** | **0.177** | **8.941** | **2.806** | **0.132** | 0.218 |

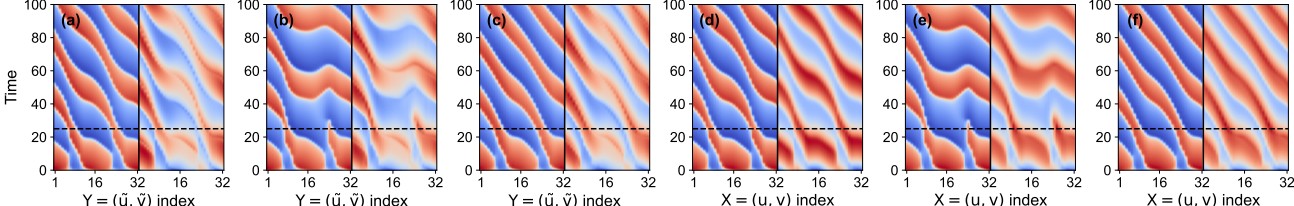

*Figure 18.* **Spatiotemporal pattern comparison on the FitzHugh–Nagumo benchmark with $\omega = 2$.** (a–c) Observation-space traveling-wave patterns for (a) ground truth, (b) CM, and (c) CM-AO. (d–f) Corresponding patterns in the canonical $(u, v)$ coordinates for (d) ground truth, (e) CM, and (f) CM-AO. The horizontal axis indexes the state components $x = (u_1, \ldots, u_{32}, v_1, \ldots, v_{32})$, while the vertical axis denotes time $t = T_i \Delta t$, where $T_i \in \{1, \ldots, 2000\}$ is the time-step index and $\Delta t = 0.05$.

## F.4. FitzHugh–Nagumo Reaction–Diffusion

Tables 5 and 6 report the invariant-set metrics and trajectory MSE for the FitzHugh–Nagumo benchmark, evaluated in the observation space and in the canonical $(u, v)$ coordinates, respectively. Both tables include the $\omega = 1$ interpolation setting and the $\omega = 2$ extrapolation setting. The proposed CM-AO performs comparably to or better than competing methods in nearly all evaluations. Notably, its gains are most pronounced in the extrapolative $\omega = 2$ setting, particularly in the canonical coordinates. These results suggest that CM-AO does not merely fit the observed trajectories, but more faithfully recovers the structure of the original FitzHugh–Nagumo dynamics by identifying a minimal topological correction through the adversarial orthogonality-based decomposition.

We visualize representative spatiotemporal rollout patterns of CM and CM-AO in the $\omega = 2$ setting. In each panel, the horizontal dotted line marks the training horizon of the target trajectory, $t_{\text{train}} = 500\Delta t = 25$. As shown in Fig. 18, both CM (b,e) and CM-AO (c,f) accurately reproduce the ground-truth traveling-wave pattern (a,d) up to $t_{\text{train}}$. Beyond the training horizon, however, a clear distinction emerges. The ground truth exhibits a diagonal traveling wave with coherent spatial phase organization, whereas CM produces a distorted pattern: temporal oscillations persist, but the diagonal wavefront structure is largely collapsed. In contrast, CM-AO successfully recovers the diagonal traveling-wave structure in the long-time regime. This suggests that CM can select a correction–gauge decomposition that is compatible with the observed training data, yet distorts the underlying canonical dynamics beyond the training horizon. By enforcing adversarial orthogonality, CM-AO instead promotes a minimal correction relative to the source dynamics, thereby selecting a more canonical decomposition.

## G. Ablation Studies

This section presents ablation studies of the proposed CM-AO framework. We compare different adversarial flow architectures, evaluate sensitivity to the regularization strength, and examine the role of the warm-up-based training curriculum. In addition to the invariant-set metrics, we report *leakage* and *alignment* for the ablation studies, following Fig. 4. Leakage, $\cos^2(\angle(\Delta f_{\mathrm{pred}}, r_{\mathrm{gt}}))$, is better when lower, whereas alignment, $\cos^2(\angle(\Delta f_{\mathrm{pred}}, \Delta f_{\mathrm{gt}}))$, is better when higher.

**Adversarial flow architecture.** We ablated the architecture used to parameterize the model-induced orbit-tangent subspace in the proposed method. In addition to the baseline Real NVP architecture in Appendix B, we considered a simpler volume-preserving NICE model (Dinh et al., 2014) and a more expressive Continuous Normalizing Flow (CNF) (Chen et al., 2018) for the adversarial flow $\Phi_\gamma$ that generates worst-case orbit-tangent directions.

As shown in Table 7, the strongly constrained NICE architecture leads to a clear performance degradation, whereas Real NVP and CNF achieve broadly comparable results. This suggests that when the adversary $\Phi_\gamma$ is substantially less expressive than the conjugacy map $\Phi_\theta$, diffeomorphic leakage can occur because the approximated orbit-tangent family is not rich enough to capture the relevant orbit directions. At the same time, the comparable per-

*Table 7.* Ablation over adversarial orbit-tangent architectures for normal-form identification in the Hopf experiment of Section 4.2.

| Metric | NICE | Real NVP | CNF |
|---|---|---|---|
| $d_H \downarrow$ | $0.093 \pm 0.030$ | $\underline{0.081 \pm 0.032}$ | $\mathbf{0.075 \pm 0.044}$ |
| $W_2 \downarrow$ | $0.081 \pm 0.024$ | $\underline{0.071 \pm 0.018}$ | $\mathbf{0.067 \pm 0.022}$ |
| MMD $\downarrow$ | $0.135 \pm 0.048$ | $\mathbf{0.109 \pm 0.036}$ | $\underline{0.124 \pm 0.076}$ |
| Leakage $\downarrow$ | $0.049 \pm 0.021$ | $\underline{0.025 \pm 0.030}$ | $\mathbf{0.021 \pm 0.037}$ |
| Alignment $\uparrow$ | $0.615 \pm 0.163$ | $\mathbf{0.780 \pm 0.133}$ | $\underline{0.756 \pm 0.189}$ |

formance of Real NVP and CNF indicates that the regularizer does not need an adversary that fully spans the entire orbit-tangent family. In practice, it appears sufficient for the adversary to capture the dominant orbit-tangent directions that overlap most strongly with the learned modulation space, which may explain the limited gain from using CNF.

**Orthogonality regularization strength.** We conducted an ablation study on the regularization strength $\lambda_{\mathrm{orth}}$ for the proposed adversarial orthogonality regularizer. As shown in Table 8, the method remains stable over a moderate range of $\lambda_{\mathrm{orth}}$, whereas excessively large values degrade performance. This degradation occurs because, in our low-data noisy regime, overly strong orthogonality regularization can bias the model toward trivial dynamics that satisfy the orthogonality constraint at the expense of the data-driven correction.

To mitigate this issue, we also considered a simple hinge variant of the orthogonality regularization,

$$\widehat{\mathcal{L}}_{\mathrm{orth}}^{\mathrm{hinge}}(W, \gamma) = \left[ \widehat{\mathcal{L}}_{\mathrm{orth}}(W, \gamma) - m_{\mathrm{orth}} \right]_+^2,$$

with $m_{\mathrm{orth}} = 0.05$. This modification is motivated by the fact that, in noisy low-data regimes, small residual alignments may not be statistically distinguishable from harmless estimation noise. Enforcing exact orthogonality can therefore make the regularizer ill-posed, causing it to penalize noise-level fluc-

*Table 8.* Ablation over $\lambda_{\mathrm{orth}}$ in the Hopf experiment of Section 4.2. Here, $\lambda_{\mathrm{orth}} = 0$ denotes the CM baseline without orthogonality regularization, and Hinge denotes the hinge-loss variant applied only to the main step.

| $\lambda_{\mathrm{orth}}$ | $d_H \downarrow$ | $W_2 \downarrow$ | MMD $\downarrow$ | Leakage $\downarrow$ | Alignment $\uparrow$ |
|---|---|---|---|---|---|
| 0 (CM) | 0.142 | 0.119 | 0.177 | 0.142 | 0.460 |
| $10^{-4}$ | 0.092 | 0.077 | 0.112 | 0.017 | 0.762 |
| $3 \times 10^{-4}$ | $\underline{0.076}$ | 0.074 | 0.115 | $\underline{0.011}$ | 0.773 |
| $10^{-3}$ | 0.081 | $\underline{0.071}$ | 0.109 | 0.025 | $\mathbf{0.780}$ |
| $10^{-3}$ (Hinge) | 0.077 | $\underline{0.071}$ | $\mathbf{0.106}$ | 0.017 | $\underline{0.776}$ |
| $3 \times 10^{-3}$ | $\mathbf{0.071}$ | $\mathbf{0.069}$ | 0.121 | $\mathbf{0.006}$ | 0.755 |
| $10^{-2}$ | 0.140 | 0.121 | 0.304 | 0.013 | 0.577 |
| $10^{-2}$ (Hinge) | 0.078 | 0.073 | $\mathbf{0.106}$ | $\underline{0.011}$ | 0.762 |

tuations rather than genuine gauge leakage. The hinge variant achieves comparable performance at moderate regularization strength, $\lambda_{\mathrm{orth}} \sim 10^{-3}$, while substantially improving performance under overly strong regularization, $\lambda_{\mathrm{orth}} \sim 10^{-2}$.

**Warm-up ablation.** We conducted an additional ablation to evaluate the sensitivity of the proposed method to the training curriculum, namely the warm-up stage with cosine ramping. As shown in Table 9, the final performance remains broadly comparable with and without warm-up, with the no-warm-up setting performing slightly better on some metrics. However, training without warm-up exhibits larger variance across runs, suggesting that the warm-up primarily improves training stability rather than the final converged performance.

*Table 9.* Effect of warm-up with cosine ramping.

| Metric | w/o warm-up | with warm-up |
|---|---|---|
| $d_H \downarrow$ | $0.076 \pm 0.048$ | $0.081 \pm 0.032$ |
| $W_2 \downarrow$ | $0.069 \pm 0.037$ | $0.071 \pm 0.018$ |
| MMD $\downarrow$ | $0.112 \pm 0.081$ | $0.109 \pm 0.036$ |
| Leakage $\downarrow$ | $0.006 \pm 0.006$ | $0.025 \pm 0.030$ |
| Alignment $\uparrow$ | $0.722 \pm 0.214$ | $0.780 \pm 0.133$ |

