# OpenReview forum: "Identifiable Smooth Conjugacy Learning via Adversarial Orthogonality"
_ICML.cc/2026/Conference — ICML 2026 regular_

### Official Review · Reviewer_1TiV · 2026-03-11

**Soundness:** 4
**Presentation:** 3
**Significance:** 3
**Originality:** 4
**Overall Recommendation:** 4
**Confidence:** 4

**Summary:**

This paper studies the problem of learning dynamical systems under limited observations by leveraging structural prior knowledge from a related source system. Classical conjugacy-based approaches assume that the source and target systems are smoothly conjugate, allowing a learned diffeomorphic transformation to transfer qualitative dynamical structure. The authors relax this assumption by introducing a low-dimensional corrective modulation of the source dynamics, parameterized via structured weight adaptation. Because jointly learning a correction and a coordinate transformation is inherently non-identifiable, the paper proposes an adversarial orthogonality constraint that encourages the learned correction to lie outside directions that can be explained by diffeomorphic reparameterization.

The resulting method aims to recover minimal topology-changing modifications to the source dynamics from sparse target observations. Experiments on bifurcation benchmarks (Hopf and Duffing systems) suggest that the proposed approach improves recovery of invariant sets, symmetry, and normal-form structure compared to several conjugacy-based baselines.

Overall, the paper focuses on a core question: how to leverage structural knowledge from a well-understood source dynamical system to identify target dynamics from sparse observations when the systems may lie in different topological regimes. The proposed framework attempts to disentangle coordinate changes from intrinsic dynamical variation to obtain interpretable corrections.

**Compliance With Llm Reviewing Policy:**

Affirmed.

**Key Questions For Authors:**

The experimental setup considers source–target pairs that lie on opposite sides of a bifurcation, but the model is still trained on trajectories from the target regime. This setting therefore appears closer to low-data system identification than to out-of-domain generalization. Could the authors clarify in what precise sense the method improves generalization, and provide additional evaluation under distributional shift (e.g., prediction from unseen initial conditions or regime-switching trajectories)?


How essential is access to a reasonably accurate source dynamical system in practice? How do purely data-driven approaches trained only on limited target data perform in these benchmarks, and under what data regimes does the proposed transfer-based approach provide clear benefits?


The experiments focus on low-dimensional bifurcation examples. Do the authors expect the adversarial orthogonality principle to remain effective in chaotic or high-dimensional systems, where invariant sets are harder to estimate and structural differences may not admit low-rank corrections?

Could the authors describe concrete empirical settings in which the assumptions of the framework — such as approximate conjugacy and low-dimensional unfolding structure — are likely to be satisfied? I.e., what real-world applications do the authors foresee for their method?


Conceptually, how far can the corrected source dynamics deviate from the original system before the structural prior becomes uninformative? Is there a principled notion of similarity between source and target systems that determines when the approach is expected to succeed?

If one were to apply this method in practice, one would have to make a choice on the source dynamics. How would this be done in practice?

**Limitations:**

No limitations section

**Strengths And Weaknesses:**

Strengths

The paper addresses an important and timely challenge at the intersection of machine learning and dynamical systems: how to exploit structural priors to recover qualitative dynamics in low-data regimes and under regime mismatch. The focus on identifiability and topology-aware learning is conceptually compelling and relevant to scientific applications.

The introduction of adversarial orthogonality as a minimality principle for separating diffeomorphic and intrinsic variations is technically interesting and grounded in dynamical-systems theory (e.g., unfolding and conjugacy concepts).

The experimental comparisons between augmentation, context modulation, and the proposed orthogonality-regularized variant help clarify the role of the constraint and provide evidence that the method learns more structured and interpretable corrections rather than relying on unconstrained fitting.

The paper goes beyond short-term prediction error and evaluates models via invariant-set discrepancies, symmetry recovery, and bifurcation structure. These metrics more directly assess whether the learned models capture long-term qualitative dynamics, which is aligned with the stated goals of the method.

Weaknesses

The introduction motivates the work in terms of extrapolation and out-of-domain behavior in phase space. However, the empirical evaluation primarily demonstrates qualitative reconstruction of target dynamics from sparse data within the observed regime. The experiments do not explicitly evaluate predictive generalization under distribution shift (e.g., unseen initial conditions, regime switching, or forecasting outside the training region), making the link to generalization somewhat indirect.

The benchmarks are limited to low-dimensional normal-form systems undergoing simple bifurcations. While appropriate as proof-of-concept tests, it remains unclear how the approach scales to more complex settings such as high-dimensional or chaotic dynamics, where invariant-set identification and structural recovery are significantly more challenging.

The proposed framework assumes access to a structurally informative source dynamical system. The empirical necessity of this assumption is not fully explored. In particular, the paper does not compare against purely data-driven models trained only on the target data to clarify when transfer from a source model provides clear advantages.

The method relies on assumptions such as low-dimensional topology-changing corrections and approximate conjugacy between source and target systems. The paper would benefit from a clearer discussion of realistic application scenarios where these assumptions are expected to hold.

All experiments appear to be synthetic. Demonstrating performance on real dynamical data would strengthen the empirical relevance of the approach.

---

> ### Author Rebuttal · Authors · 2026-03-31
>
> We sincerely appreciate the reviewer’s constructive feedback. Below, we address each comment carefully. Additional results are available in the **README** at https://github.com/anonymous365954/s24252 and will be thoroughly incorporated into our revised paper.
>
> ***
>
> **1. A more challenging benchmark**
>
> To test the framework in a higher-dimensional setting, we applied it to a reaction-diffusion PDE, spatially discretized into a 64D ODE in the $(u,v)$ state space. The source model was trained on abundant trajectories from the $\mu = 0$ quiescent resting regime, where solutions converge to a homogeneous fixed point. We then transferred this source model using a single noisy target trajectory from the $\mu = 0.6$ active oscillatory regime, under a nonlinear observation distortion.
>
> We evaluated each method using both 64D invariant-set metrics and a qualitative visualization test of long-horizon traveling-wave reproduction. As shown in **Table 1 and Figure 1** at the provided link, the proposed CM-AO outperforms the competing baselines; notably, in the extrapolative setting, it is the only method that preserves a clean, non-blow-up traveling-wave pattern.
>
> ***
>
> **2. Target-only baseline**
>
> We conducted the Hopf experiment in a purely target-only setting, where a NODE is trained directly on target observations without any source prior. As shown in **Table 4** at the provided link, even with $N_t = 5$ trajectories, the target-only baseline achieves at best on par with CM-AO using only $N_t = 1$. This highlights a key advantage of conjugacy-based transfer: by leveraging source structural priors, it recovers the underlying canonical dynamics from limited target data.
>
> ***
>
> **3. Generalization under distributional shift**
>
> We agree that the current draft did not clearly distinguish different notions of generalization. Our main focus is parameter-level generalization: Figure 7 in the manuscript shows that CM-AO recovers the correct bifurcation structure beyond the interpolation region. To address the reviewer’s point on initial-condition generalization, we revisited the Hopf experiment under explicit initial-condition extrapolation. Specifically, we restricted the target trajectories to initial conditions far from the limit cycle and used a short time horizon, so that the training data did not reveal the limit-cycle geometry. As shown in **Table 5** at the provided link, the proposed CM-AO outperforms all competing methods. This suggests that the framework goes beyond fitting short trajectories, recovering the canonical structure and the correct limit cycle from off-limit-cycle initializations.
>
> ***
>
> **4. Source-target similarity**
>
> A principled way to view source-target similarity is through unfolding theory: the target vector field $g$ should lie close to the orbit of an unfolding of $f_0$, i.e., $g\approx \Phi_*(f_0+\sum_i \xi_i p_i)$. Accordingly, if target trajectories cannot be reconstructed well from $f_0$ using only a small context modulation with a diffeomorphic map, this suggests that the chosen source is structurally mismatched. From this, two practical diagnostics are informative: the conjugacy loss itself and the norm/sparsity of the weights of modulation basis $P_W$, i.e., $\\|W\\|_p$. If neither decreases during training, this indicates that even a complex modulation basis cannot yield a good conjugate reconstruction, implying that the chosen source is not close enough to the target dynamics.
>
> To assess this, we evaluated CM-AO on the Hopf benchmark using a harmonic oscillator as a structurally mismatched source system. As shown in **Figure 2** at the provided link, under this mismatch, the conjugacy loss and, in particular, the basis-weight sparsity (group-lasso norm) do not decrease but instead plateau across epochs. This provides a practical indication that the chosen source is not a suitable prior.
>
> ***
>
> **5. Practical application and source selection**
>
> A realistic use case for our method is transfer across families of physical systems that share a common qualitative structure but differ by mild parameter drift or observation distortion. Examples include oscillatory or resonant systems, where a canonical source captures the main dynamical backbone, while individual target instances deviate due to fabrication variability or operating-condition shifts. Our method targets this regime by using the source as a structural anchor and modeling each target through a low-dimensional intrinsic correction together with removable diffeomorphic variation.
>
> Accordingly, the source should be chosen to capture the broad qualitative behavior of the phenomenon of interest, ideally in the form of a reduced model for a representative regime. When such a hand-crafted model is unavailable, the source can instead be a neural surrogate trained on a well-observed parameter slice, which then serves as the reference for transfer to nearby regimes. This enables automatic extension of a scientific model across parameter space.

---

> > ### Author Rebuttal · Reviewer_1TiV · 2026-04-03
> >
> > I thank the authors for thoroughly addressing most of my points. They're results are convincing and I want to emphasize that their method has a number of interesting and appealing aspects. Nonetheless, it appears that the use cases are overly specific, and in my opinion partly unrealistic, such that the application field in practice is very limited. Due to this, I believe my Initial score is warranted.

---

> > > ### Author Response · Authors · 2026-04-04
> > >
> > > Thank you for taking the time to review the additional results and for noting that our rebuttal addressed the main concerns. We also sincerely appreciate your positive assessment of the method’s technical aspects.
> > >
> > > We understand the concern that the present use cases may appear somewhat specialized. Our intention, however, is not to position the method as a universally applicable framework for all dynamical modeling problems, but rather as a principled approach for a focused class of settings in which recovering the underlying canonical structure is especially important. We believe such settings remain practically meaningful in scientific modeling, as properties such as topological compatibility, invariant-set recovery, bifurcation behavior, and symmetry-related structure can play an important role in reliable prediction, extrapolation, and interpretation.
> > >
> > > We agree that this scope should be articulated more explicitly, and we are grateful for your comment in helping us clarify that framing. Thank you again for your careful reading and constructive feedback.

---

### Official Review · Reviewer_qxvh · 2026-03-12

**Soundness:** 3
**Presentation:** 4
**Significance:** 3
**Originality:** 3
**Overall Recommendation:** 4
**Confidence:** 4

**Summary:**

This work is about a core question in dynamical systems modeling ie, how to handle topological mismatches between a source model and target data. In conjugacy learning, the model is warped via a diffeomorphism to match the data. This process becomes ill posed in case the source and target are not topologically equivalent. The authors examine this by proposing a method that learns a low-rank correction alongside the warp. Adversarial orthogonality is used as a constraint to ensure the correction doesn't leak into other directions.

**Compliance With Llm Reviewing Policy:**

Affirmed.

**Key Questions For Authors:**

You define identifiability relative to the chosen model class. Have you tested the sensitivity of the clean dynamics to the architecture of the adversarial normalizing flow?

If the adversary is significantly simpler than the conjugacy map , does diffeomorphic leakage reappear?

While Theorem A.1 provides local intuition is there a formal way to proving that the low rank modulation is guaranteed to recover a transversal slice in the presence of neural approximation errors?

How do you expect the adversarial orthogonality to scale to systems where the orbit-tangent space is significantly larger?

**Limitations:**

To me, the paper as a concept is strong , and I find the geometric approach to deal with conjugacy very promising. However, the gap between the claims of identifiability and the toy scale evidence is currently too large. I see this as a real limitation. More to this, the theory is mainly first-order and the experiments are restricted to 2D. I suggest to expand on the connection between the neural implementation and the unfolding theory or providing results on higher-dimensional systems, so the paper becomes more impactful.

**Strengths And Weaknesses:**

The paper identifies an ambiguity in conjugacy based learning. The presentation of the paper is a bit dense. I think the examples and the intuition are not hard to understand the way it is presented though. I will start with what I consider as the strengths of this work. I find the geometric motivation which uses Lie brackets to define orbit tangent directions elegant and well-grounded in the dynamical systems theory. The derivation of orbit tangent directions in Section 3.3 (Definition 3.1 and Lemma 3.2) is precise. The connection to VUT in Apx. A provides a nice justification for why first-order orthogonality is a sufficient proxy for capturing intrinsic changes. The experiments on Hopf and Duffing systems are well chosen. The recovery of the hidden symmetry in Figure 6 is a compelling demonstration for the method. Also, the authors provide an alternative to the single direction alignment in Apx D. I also found strenght in the worst case trace ratio.

However, I found some unsupported/not well backed claims throughout the paper. For instance, to provide an identifiable decomposition (e.g., Lines 13–18), this identifiability is relative to the chosen model induced tangent family (Section 3.4). The paper lacks a formal proof or a more generic/hard empirical study to show that this separation holds beyond the specific adversarial architecture used. While the 2D benchmarks are illustrative, they are essentially best-case scenarios and simple. It remains unclear to me how the adversarial scan (Section 3.4) performs when the orbit tangent space becomes high-dimensional or when the diffeomorphism group is more complex. The method relies on a specific training curriculum too. The sensitivity of the final/clean dynamics to these settings needs more exploration. How the penalty affects the decomposition? Does it suppress topological corrections otherwise valid? And the sufficiency claim (Lines 289–293) relys on the adversary being expressive enough to capture all relevant diffeomorphic variations. The paper does not analyze what happens when the tangent family is restricted. I think this is a key point for an identifiable framework.

---

> ### Author Rebuttal · Authors · 2026-03-31
>
> We sincerely appreciate the reviewer’s constructive feedback. Below, we address each comment carefully. Additional results are available in the **README** at https://github.com/anonymous365954/s24252 and will be thoroughly incorporated into our revised paper.
>
> ***
>
> **1. A more challenging benchmark**
>
> To test the framework in a higher-dimensional setting, we applied it to a reaction-diffusion PDE, spatially discretized into a 64D ODE in the $(u,v)$ state space. The source model was trained on abundant trajectories from the $\mu = 0$ quiescent resting regime, where solutions converge to a homogeneous fixed point. We then transferred this source model using a single noisy target trajectory from the $\mu = 0.6$ active oscillatory regime, under a nonlinear observation distortion.
>
> We evaluated each method using both 64D invariant-set metrics and a qualitative visualization test of long-horizon traveling-wave reproduction. As shown in **Table 1 and Figure 1** at the provided link, the proposed CM-AO outperforms the competing baselines; notably, in the extrapolative setting, it is the only method that preserves a clean, non-blow-up traveling-wave pattern.
>
> ***
>
> **2. Sensitivity to the orbit-tangent architecture**
>
> We performed an ablation study comparing different architectures of the model-induced orbit-tangent directions. Beyond the baseline affine coupling (RNVP), we evaluated a simpler volume-preserving architecture (NICE) and a more expressive neural vector field (CNF). **Table 2** at the provided link shows that the strongly constrained NICE leads to a clear degradation, while RNVP and CNF exhibit broadly comparable performance. As the reviewer suggested, this indicates that when the adversary is much less expressive than the conjugacy map, diffeomorphic leakage can emerge since the approximated orbit-tangent family is no longer rich enough to capture the relevant orbit directions. At the same time, the results suggest that, in practice, the regularizer does not require an adversary that fully spans the entire orbit-tangent family. Rather, it appears sufficient for the adversary to capture the dominant realizable orbit-tangent directions that overlap most strongly with the learned modulation space $P_W$, which likely explains the limited gain from CNF.
>
> ***
>
> **3. Does the orthogonality penalty suppress topological corrections?**
>
> The orthogonality regularizer is designed to suppress the part of the learned correction that is realizable within the diffeomorphic family, not to remove intrinsic corrections altogether. When the ground-truth mismatch consists of an unavoidable topological correction plus a diffeomorphic gauge, the regularizer drives the model to discard the gauge-explainable part while preserving the correction required for data fitting, since eliminating the latter would increase the reconstruction (conjugacy) error.
>
> However, the regularizer can indeed become harmful when its weight is too large relative to the conjugacy objective. In that case, the model may favor trivial dynamics that satisfies the orthogonality at the expense of the correction required by the data. This effect is particularly pronounced in our present regime, where only a small number of noisy target trajectories are available. To examine this, we conducted an ablation study over $\lambda_{orth}$, reported in **Table 3** at the provided link. The results show that the method remains stable over a moderate range of $\lambda_{orth}$, whereas excessively large values lead to performance degradation.
>
> To mitigate this issue, we further considered a simple hinge loss variant of the orthogonality loss. This modification is motivated by the observation that, in low-data noisy regimes, enforcing exact orthogonality may over-bias the model under strong regularization. The hinge yields comparable performance at the moderate $\lambda_{orth} \sim 10^{-3}$, while substantially improving performance at the overly large $\lambda_{orth} \sim 10^{-2}$.
>
> ***
>
> **4. Sensitivity to the training curriculum**
>
> We also evaluated sensitivity of the proposed method to warm-up training. **Table 6** at the provided link shows that final performance is broadly similar with and without warm-up, with no warm-up even slightly improving some metrics. At the same time, training without warm-up tends to exhibit larger variance across runs, suggesting that the warm-up primarily improves training stability.
>
> ***
>
> **5. On formal guarantees**
>
> We thank the reviewer for this important question. At present, we do not claim a formal guarantee that the learned low-rank modulation recovers a transversal slice in the presence of approximation errors. Theorem A.1 is a local existence result for versal unfolding in an idealized setting, and the proposed method should be understood as a practical surrogate that encourages approximate transversality relative to a model-induced orbit-tangent family. We will explicitly acknowledge this limitation in the revised paper.

---

> > ### Author Rebuttal · Reviewer_qxvh · 2026-04-01
> >
> > I followed the link and overall the rebuttal's effort and results made the paper more convincing to me. The new higher-dimensional reaction-diffusion example helps with my concern that the original evidence was limited to very simple 2D cases. I also found the ablation on the adversarial architecture and the sensitivity checks on the orthogonality weight and warm-up useful, since they speak directly to some of the practical questions I had about the method. I still think the paper should be a bit more careful with the term “identifiability.” The method still seems to give a relative, model-dependent notion of separation, not a formal guarantee. That said, I changed my score (3->4) wrt the needs of a revision.

---

> > > ### Author Response · Authors · 2026-04-04
> > >
> > > Thank you for taking the time to review the additional results and for noting that the rebuttal made the paper more convincing. We are pleased that the new higher-dimensional reaction-diffusion example and the ablation studies helped clarify the practical questions of the method.
> > >
> > > We also appreciate your point regarding the use of the term “identifiability.” As discussed in our rebuttal, we intend this as a model-relative, approximated notion of local decomposition, rather than a fully general formal guarantee. We agree that this distinction should be stated more carefully, and we will revise the wording accordingly to better reflect the scope of the claim.
> > >
> > > Thank you again for the careful reading and constructive feedback.

---

### Official Review · Reviewer_tvtY · 2026-03-16

**Soundness:** 3
**Presentation:** 3
**Significance:** 2
**Originality:** 2
**Overall Recommendation:** 4
**Confidence:** 3

**Summary:**

The paper introduces a framework to improve dynamical system modeling. The central idea is to relax recent approaches based on conjugacy learning that learn diffeomorphic maps that push forward a source vector field by adding/allowing for low-rank context-modulated correction the source dynamics. A key innovation is an adversarial orthogonality constraint: the correction is forced to be orthogonal to orbit‑tangent directions induced by the worst‑case diffeomorphisms. This prevents the diffeomorphism and the intrinsic dynamical correction from collapsing into non‑identifiable trade‑offs.

**Compliance With Llm Reviewing Policy:**

Affirmed.

**Final Justification:**

I thank the authors for their detailed and thoughtful responses. In light of these clarifications and the additional evidence provided, I find the contribution clearer and stronger than before. I have updated my score accordingly and now recommend the paper for acceptance.

**Key Questions For Authors:**

- It is not entirely clear to me why this method would result in identifiable class. Since "identifiable" is even in the title, it would help if the meaning of identifiability in this context could be made more precise, and ideally claims could be made more rigorously.
- What is the computational cost / timings
- How are diffeomorphism \Phi_\gamma parameterized? What are design considerations here, especially since it seems this part if thrown away after training.
- When do we expect the orthogonality regularization to be beneficial and in what settings may it hurt?

**Limitations:**

yes

**Strengths And Weaknesses:**

Strengths:
1. Clear motivation and strong conceptual grounding (convincing argument that existing conjugacy-learning method can be overly constraint if source/target are not topologically compatible), and that overcoming such misspecification (own interpretation) can improve generalization.
2. The adversarial orthogonality mechanism seems like an interesting contribution to solve the problem.
3. Use of low‑rank modulation seems like a simple and natural choice as baseline to assess whether a method may benefit from corrections altogether.

Weaknesses:
- Adversarial search over diffeomorphisms plus integration of modulated NODEs may be expensive. Runtime estimates, scaling analyses, or approximations would be welcome.
- Reliance on GANs (added complexity, can be harder/unstable to train, and require discriminator that becomes redundant after training)

---

> ### Author Rebuttal · Authors · 2026-03-31
>
> We sincerely appreciate the reviewer’s constructive feedback. Below, we address each comment carefully. Additional results are available in the **README** at https://github.com/anonymous365954/s24252 and will be thoroughly incorporated into our revised paper.
>
> ***
>
> **1. The meaning of identifiability**
>
> Our intended notion is a local first-order identifiability of the discrepancy between a target system $g$ and the reference $f_0$, decomposed into (i) diffeomorphic variation along the orbit of $f_0$, and (ii) intrinsic, topology-changing variation transverse to that orbit. To make this precise, we will add the following definition to the revised manuscript:
>
> Let $T := T_{f_0} O_{f_0}$ be the orbit-tangent space at $f_0$, and let $P$ be a candidate intrinsic modulation space. We say that $P$ provides a locally identifiable first-order representation of the intrinsic correction modulo diffeomorphic variation, if every first-order discrepancy $h \in T + P$ admits a unique decomposition $h = t + p$ where $t \in T$ and $p \in P$.
>
> Here $h$ represents the target discrepancy of $g$ relative to $f_0$, i.e., $h = g - f_0$. This decomposition is unique whenever $P \cap T= \\{0\\}$, and in particular when $P \perp T$ under the chosen metric. This is the motivation for the orthogonality regularization proposed in the paper.
>
> ***
>
> **2. Design consideration of $\Phi_\gamma$**
>
> Our baseline choice is to parameterize the adversary $\Phi_\gamma$ using the same class as the conjugacy map $\Phi_\theta$. The adversary is meant to identify and suppress orbit-tangent directions that are realizable by the model diffeomorphism, so matching this class provides a consistent baseline. Particularly, we use affine couplings since they are expressive enough to model nontrivial maps, while permitting low-cost computation of the associated generators.
>
> To examine the sensitivity of our method to this architectural choice, we performed an ablation study comparing different parameterizations of $\Phi_\gamma$. Beyond the baseline affine coupling (RNVP), we evaluated a simpler volume-preserving architecture (NICE) and a more expressive neural vector field (CNF). **Table 2** at the provided link shows that the strongly constrained NICE parameterization leads to a clear degradation in performance, while RNVP and CNF exhibit broadly comparable performance. This indicates that when the adversary is much less expressive than the conjugacy map, diffeomorphic leakage can emerge since the approximated orbit-tangent family is no longer rich enough to capture the relevant orbit directions. At the same time, the results suggest that, in practice, the regularizer does not require an adversary that fully spans the entire orbit-tangent family. Rather, it appears sufficient for the adversary to capture the dominant realizable orbit-tangent directions that overlap most strongly with the learned modulation space $P_W$, which likely explains the limited gain from CNF.
>
> ***
>
> **3. When orthogonality helps or hurts**
>
> The orthogonality regularizer is beneficial in the regime assumed by our formulation: when the ground truth mismatch consists of an unavoidable topological correction together with a realizable diffeomorphic gauge. In this setting, the regularizer suppresses the gauge-explainable part, while retaining the topological correction that is required to fit the data: since removing the intrinsic correction would increase the reconstruction (conjugacy) error, the optimization is driven toward a decomposition that is as transverse as possible.
>
> By contrast, the regularizer can become harmful when its weight is too large relative to the conjugacy objective. In that case, the model may favor trivial dynamics that satisfies the orthogonality at the expense of the correction required by the data. This effect is particularly pronounced in our present regime, where only a small number of noisy target trajectories are available. To examine this, we conducted an ablation study over $\lambda_{orth}$, reported in **Table 3** at the provided link. The results show that the method remains stable over a moderate range of $\lambda_{orth}$, whereas excessively large values lead to performance degradation.
>
> To mitigate this issue, we further considered a simple hinge loss variant of the orthogonality loss. This modification is motivated by the observation that, in low-data noisy regimes, enforcing exact orthogonality may over-bias the model under strong regularization. The hinge yields comparable performance at the moderate $\lambda_{orth} \sim 10^{-3}$, while substantially improving performance at the overly large $\lambda_{orth} \sim 10^{-2}$.
>
> ***
>
> **4. Computational cost**
>
> We report the wall-clock time and number of trainable parameters in **Table 7** at the provided link. Due to adversarial learning, CM-AO uses additional parameters and requires roughly $2.2\times$ more computation time. We will state this limitation explicitly in the revised manuscript.

---

> > ### Author Rebuttal · Reviewer_tvtY · 2026-04-07
> >
> > I thank the authors for their detailed and thoughtful responses. In light of these clarifications and the additional evidence provided, I find the contribution clearer and stronger than before. I have updated my score accordingly and now recommend the paper for acceptance.

---

> > > ### Author Response · Authors · 2026-04-07
> > >
> > > Thank you for taking the time to review the additional results and for your thoughtful follow-up response. We are greatly encouraged that the clarifications and additional evidence helped make the contribution clearer and stronger.
> > >
> > > Thank you again for your careful reading and constructive feedback.

---

### Decision · Program_Chairs · 2026-04-30

**Decision:**

Accept (regular)

**Comment:**

This paper studies the problem of learning a dynamical system from observations.  The context here is that one is learning the dynamical system with guidance from a related system.  One important consideration is that of preserving the topological invariance of the system.  A class of methods that achieve this are conjugacy-based methods; however, these methods do assume that the source and target systems are conjugate in the first place.  This work relaxes such an assumption by introducing a low-rank correction.  To deal with identifiability issues, the authors introduce an adversarial orthogonality constraint so that the learned correction lies in a space not due to diffeomorphic reparameterization.

The reviewers unanimously agreed on the positive aspects of the paper; namely that the problem is important and timely, and that the solution makes sense from a geometric as well as a computational viewpoint.  On these, I am happy to recommend acceptance to the conference.

The reviewers did also provide a list of very constructive feedback to help improve the paper (there were concerns about how applicable the method was and the lack of real-life experiments).  I think these will significantly strengthen the ideas in these paper, and urge the authors take the opportunity to address these.